# Open Problems and Fundamental Limitations of Reinforcement Learning from Human Feedback

**Stephen Casper,**[*] *MIT CSAIL,* `scasper@mit.edu`
**Xander Davies,**[*] *Harvard University*

**Claudia Shi,** *Columbia University*
**Thomas Krendl Gilbert,** *Cornell Tech*
**Jérémy Scheurer,** *Apollo Research*
**Javier Rando,** *ETH Zurich*
**Rachel Freedman,** *UC Berkeley*
**Tomasz Korbak,** *University of Sussex*
**David Lindner,** *ETH Zurich*
**Pedro Freire,** *Independent*
**Tony Wang,** *MIT CSAIL*
**Samuel Marks,** *Harvard University*
**Charbel-Raphaël Segerie,** *EffiSciences*
**Micah Carroll,** *UC Berkeley*
**Andi Peng,** *MIT CSAIL*
**Phillip Christoffersen,** *MIT CSAIL*
**Mehul Damani,** *MIT CSAIL*
**Stewart Slocum,** *MIT CSAIL*
**Usman Anwar,** *University of Cambridge*
**Anand Siththaranjan,** *UC Berkeley*
**Max Nadeau,** *Harvard University*
**Eric J. Michaud,** *MIT*
**Jacob Pfau,** *New York University*
**Dmitrii Krasheninnikov,** *University of Cambridge*
**Xin Chen,** *ETH Zurich*
**Lauro Langosco,** *University of Cambridge*
**Peter Hase,** *UNC Chapel Hill*

**Erdem Bıyık,** *University of Southern California*
**Anca Dragan,** *UC Berkeley*
**David Krueger,** *University of Cambridge*
**Dorsa Sadigh,** *Stanford University*
**Dylan Hadfield-Menell,** *MIT CSAIL*

**Reviewed on OpenReview:** *https://openreview.net/forum?id=bx24KpJ4Eb*

## Abstract

Reinforcement learning from human feedback (RLHF) is a technique for training AI systems to align with human goals. RLHF has emerged as the central method used to finetune state-of-the-art large language models (LLMs). Despite this popularity, there has been relatively little public work systematizing its flaws. In this paper, we (1) survey open problems and fundamental limitations of RLHF and related methods; (2) overview techniques to understand, improve, and complement RLHF in practice; and (3) propose auditing and disclosure standards to improve societal oversight of RLHF systems. Our work emphasizes the limitations of RLHF and highlights the importance of a multi-layered approach to the development of safer AI systems.

---

[*]Equal contribution. Correspondence to `scasper@mit.edu`.

# Contents

# 1    Introduction

*Reinforcement learning from human feedback* (RLHF) has emerged as a prominent technique to adapt machine learning models to difficult-to-specify goals (Christiano et al., 2017; Ziegler et al., 2019; Bai et al., 2022a). In particular, RLHF is a key component of training state-of-the-art large language models (LLMs), such as OpenAI's GPT-4 (OpenAI, 2023), Anthropic's Claude2 (Anthropic, 2023), Google's Bard (Google, 2023), and Meta's Llama 2 (Touvron et al., 2023). RLHF and similar methods allow LLMs to go beyond modeling the distribution of their training data, and adapt the distribution of text so that model outputs are rated more highly by human evaluators.

We use "RLHF" to refer to methods that combine three interconnected processes: human feedback collection, reward modeling, and policy optimization. Figure 1 (top) illustrates this setup. The feedback process elicits evaluations of model outputs from humans. The reward modeling process uses supervised learning to train a reward model that imitates these evaluations. The policy optimization process optimizes the AI system to produce outputs that recieve favorable evaluations from the reward model. When it works well, RLHF leverages the relative ease of identifying 'good' behavior compared to demonstrations, manually-engineered reward functions, or other methods of specifying or learning rewards.

RLHF has its roots in revealed preference theory from economics. Revealed preference theory formalizes the idea that one can learn about an actor's goals from their behavior (Chambers and Echenique, 2016). It was adopted by the machine learning field early on for applications in human-computer interaction and reinforcement learning (Knox and Stone, 2008; 2010; 2012; Akrour et al., 2012; Fürnkranz et al., 2012; Griffith et al., 2013). Wirth et al. (2017) offers a survey of these methods. The standard methodology for RLHF used today was popularized in 2017 by Christiano et al. (2017), which has played a key role in directing the attention of the deep reinforcement learning community to feedback-based methods.

RLHF has emerged as the primary strategy to finetune LLMs before deployment (OpenAI, 2023; Anthropic, 2023; Google, 2023; Touvron et al., 2023), with the goal of producing safe models aligned with human objectives. Despite this, deployed models finetuned with RLHF have revealed sensitive private information (Li et al., 2023b; El-Mhamdi et al., 2022), hallucinated untrue content (Ji et al., 2023; OpenAI, 2023; Zhang et al., 2023; Huang et al., 2023), spread biases that favor specific political ideologies (Santurkar et al., 2023; Perez et al., 2022b), exhibited sycophantic responses (Perez et al., 2022b; Sharma et al., 2023), expressed undesirable preferences (e.g., not wanting to be shut down) (Perez et al., 2022b), and been easy to misalign by finetuning on as few as 10 new examples (Yang et al., 2023; Qi et al., 2023; Lermen et al., 2023; Zhan et al., 2023). RLHF has also not made models robust to adversarial attacks from jailbreaking (i.e., subverting the constraints the system is normally meant to operate under) or prompt injection/extraction (Choi et al., 2022; Willison, 2023; Albert, 2023; Oneal, 2023; Li et al., 2023b; Wolf et al., 2023; Liu et al., 2023; Rao et al., 2023; Wei et al., 2023; Shen et al., 2023; Shah et al., 2023).

Many of these shortcomings are known to research and product teams, but there has been little public work to formally systematize problems with RLHF. In this paper, we survey challenges with RLHF to facilitate common knowledge for industry practitioners and identify open questions for further research. We focus primarily on RLHF with LLMs due to their prominence in state-of-the-art applications. We make three contributions:

1. **Concrete challenges with RLHF:** In Section 3, we taxonomize and survey problems associated with RLHF. We divide them into three primary categories: challenges with the **human feedback**, challenges with the **reward model**, and challenges with the **policy**. We also distinguish between challenges with RLHF that are relatively **tractable** and could be addressed within the RLHF framework using improved methodology versus **fundamental** limitations of RLHF, which will require alternative approaches to address.[1]

2. **Incorporating RLHF into a broader technical safety framework:** In Section 4, we discuss how RLHF is not a complete framework for developing safe AI and highlight additional approaches

---

[1]We use color only to highlight topics. This paper can be viewed in grayscale.

that can help to better understand, improve, and complement it. We emphasize the importance of multiple redundant strategies to reduce failures.

3. **Governance and transparency:** In Section 5, we consider the challenge of improving norms and regulations affecting models trained with RLHF. Specifically, we discuss how the disclosure of certain details by companies using RLHF to train AI systems can improve accountability and auditing.

Right now, RLHF functions both as a basic technique that can be used to study AI alignment and as a practical method to align deployed systems. Here, we focus on the possibilities and limitations of the latter. However, our larger goal is to support concerted efforts to critically examine the relationship between RLHF as an alignment strategy and RLHF as an engineering tool. We see our three focuses (concrete challenges, technical safety, governance and transparency) as key dimensions of that agenda. Policymakers and researchers should invest in this type of work even as specific technical claims are superseded by future developments.

## 2 Background and Notation

### 2.1 High-Level Overview

One simple approach to training aligned AI systems would be to have humans directly evaluate AI system outputs, and train with these human evaluations as a reward signal. However, because the number of training episodes required by AI systems is large, this basic approach tends to be slow and costly. Reinforcement learning from human feedback (RLHF) provides a scalable alternative. RLHF involves three key steps: (1) collecting **human feedback** on examples of an AI system performing tasks; (2) using the feedback to train a **reward model** which imitates human evaluations; and (3) finetuning the AI system's **policy** with reinforcement learning (RL) using rewards provided by the reward model. In practice, RLHF is performed iteratively by repeating these steps (or performing them synchronously). The overall procedure is illustrated in Figure 1 (top).

RLHF can be used for many tasks, and much of the historical research with it has involved agents learning to act in simple gridworld or robotic control environments. Here, however, we focus primarily on the example of language models that generate text such as OpenAI's GPT-4 (OpenAI, 2023), Anthropic's Claude2 (Anthropic, 2023), Google's Bard (Google, 2023), and Meta's Llama 2 (Touvron et al., 2023). These models are not trained only with RLHF. Instead, they are pretrained to imitate web text before being finetuned with RLHF. In practice, RLHF finetuning for chat-based language models tends to rely heavily on obtaining binary feedback from humans on conversation pairs. This process is depicted in Figure 2. Humans are presented with pairs of conversations from the model and asked to indicate which, if any, was better than the other. Then the reward model is trained to assign scalar rewards consistent with these comparisons, and the language model is fine-tuned using RL to generate text to which the reward model assigns high reward.

### 2.2 Formal Framework

Here, we present a simple formal framework for RLHF based, in part, on the one from Christiano et al. (2017). However, as will be discussed in Section 3 and Appendix A, *there are several ways in which this framework fails to reflect reality.*

**Step 0, (Optional) Pretraining:** RLHF begins with an initial base model $\pi_\theta$ with parameters $\theta$ which generates a distribution of examples. The goal of this step is to begin with a base model that has general knowledge which can later be utilized for the target task. For example, when performing RLHF with LLMs, the base model is typically a language generator pretrained on web text.

**Step 1, Collecting human feedback:** The first step is to obtain examples (e.g. responses or conversations) from the base model and collect human feedback on those examples. The end result is a dataset of examples annotated with information about their desirability. Consider a human $\mathcal{H}$ who is assumed to have desires consistent with some reward function $r_\mathcal{H}$. A dataset of examples is sampled from $\pi_\theta$ where each example $x_i$ is defined to be a batch of one or more generations from the base model. Let the feedback function $f$ map

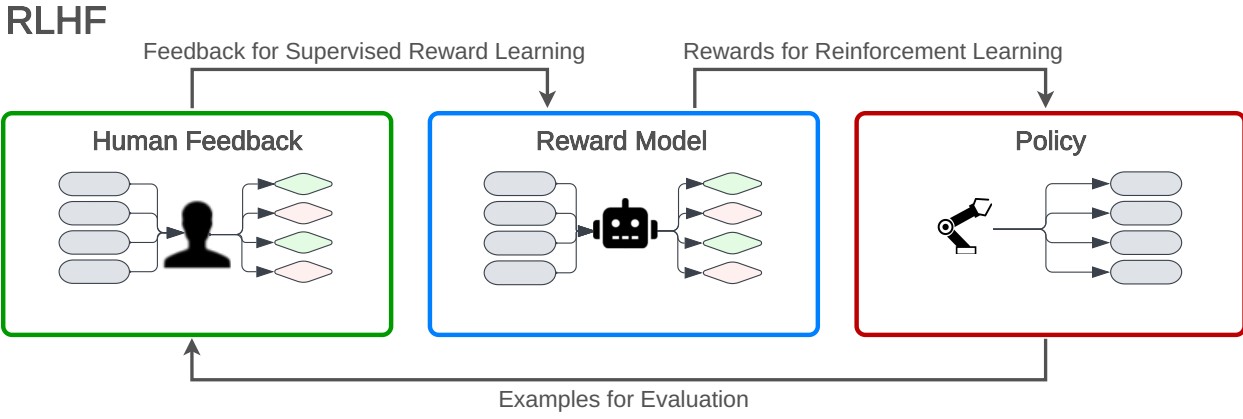

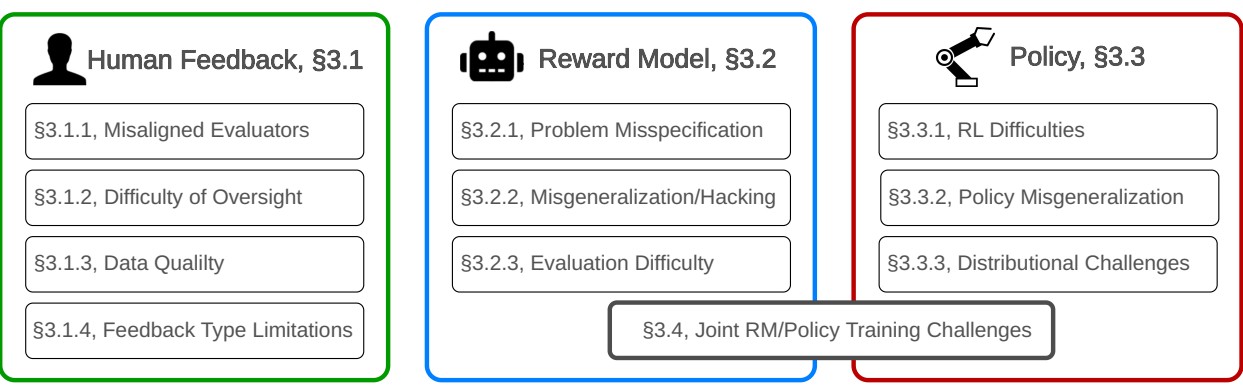

Figure 1: **(Top) Reinforcement Learning from Human Feedback.** Gray, rounded boxes correspond to outputs (e.g., text), and colored diamonds correspond to evaluations. The key steps are to (1) collect **human feedback** on examples of an AI system performing tasks, (2) use supervised learning to train a **reward model** on the human feedback data so that it can serve as a proxy for human evaluations, and (3) finetune the **policy** using reinforcement learning using the reward model's rewards. **(Bottom) Our taxonomy for challenges with RLHF.** We divide challenges with RLHF into three main types: challenges with obtaining quality **human feedback**, challenges with learning a good **reward model**, and challenges with **policy** optimization. In the figure, each contains boxes corresponding to the subsections of Section 3.

the example $x_i$ and random noise $\epsilon_i$ to feedback $y_i$. The data collection process is thus often modeled as:

$$x_i \sim \pi_\theta, \qquad y_i = f(\mathcal{H}, x_i, \epsilon_i). \tag{1}$$

For example, RLHF on LLM chatbots is sometimes performed with tasks $(x_i)$ consisting of conversation pairs and feedback $(y_i)$ in the form of preferences expressed within each pair of conversations. We survey challenges with obtaining human feedback in Section 3.1. See also Appendix A for an improved framing of the feedback process which corrects several ways in which this framing is misspecified.

**Step 2, Fitting the reward model:** The second step of RLHF is to fit a reward model $\hat{r}_\phi$ using the dataset obtained in the first step to approximate evaluations from the human $\mathcal{H}$ as closely as possible. Given a dataset of examples and preferences $\mathcal{D} = \{(x_i, y_i)_{i=1,\dots,n}\}$, the parameters $\phi$ are trained to minimize

$$\mathcal{L}(\mathcal{D}, \phi) = \sum_{i=1}^{n} \ell(\hat{r}_\phi(x_i), y_i) + \lambda_r(\phi), \tag{2}$$

## Example: LLM Chatbot RLHF from Binary Preference Feedback

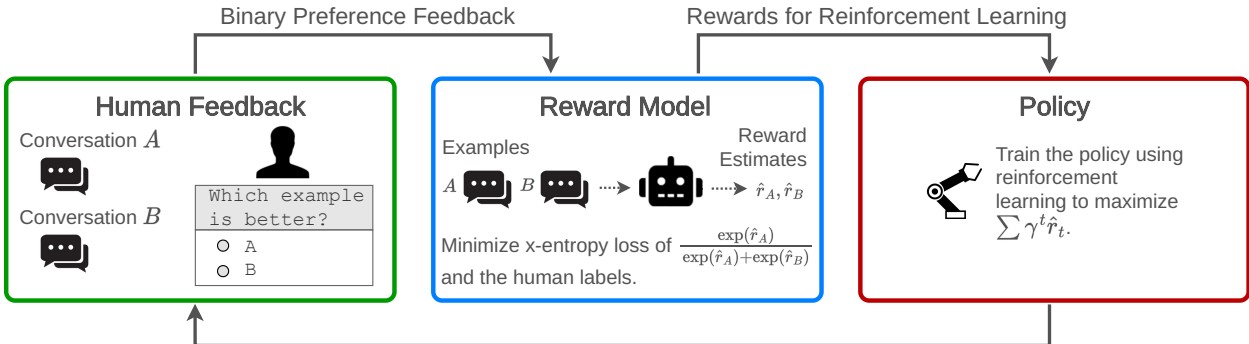

Figure 2: **An example of RLHF for finetuning chatbots with binary preference feedback.** (Left) Humans indicate which conversation between a pair they prefer. Many of these examples are collected to obtain a dataset of conversation pairs annotated with human preferences. (Middle) A reward model is trained using each example pair to provide rewards that reflect the human's decisions as closely as possible. (Right) Finally, the language model policy is finetuned using reinforcement learning. The reward signal comes from the reward model which acts as a proxy for the human.

where $\ell$ is a suitable loss function and $\lambda_r$ is some regularizer. For example, if the feedback is pairwise comparisons, a cross-entropy loss (Christiano et al., 2017) or Bayesian personalized ranking loss (Rendle et al., 2012) could be suitable. We survey challenges with reward modeling in Section 3.2.

**Step 3, Optimizing the Policy with RL:** The third and final step of RLHF is to use the reward model $\hat{r}_\phi$ as a proxy for human oversight to finetune the base model. This is done using using reinforcement learning where the model is trained to maximize rewards from the reward model. The new parameters $\theta_{\text{new}}$ of $\pi$ are trained to maximize

$$\mathcal{R}(\theta_{\text{new}}) = \mathbb{E}_{x \sim \pi_{\theta_{\text{new}}}} \left[ \hat{r}_\phi(x) + \lambda_p(\theta, \theta_{\text{new}}, x) \right], \tag{3}$$

where $\lambda_p$ is some regularizer such as a divergence-based penalty between two distributions (Korbak et al., 2022b). We survey challenges with policy optimization in Section 3.3.

**Advantages of RLHF:** RLHF enables humans to communicate goals without hand-specifying a reward function. As a result, it can mitigate reward hacking relative to hand-specified reward functions and makes reward shaping natural and implicit. RLHF also leverages human judgments, which can be easier to provide than demonstrations. These advantages have made RLHF useful for helping policies learn intricate solutions in control environments (Christiano et al., 2017; Biyik, 2022; Lee et al., 2021; Hejna and Sadigh, 2022) and for finetuning LLMs (Bai et al., 2022a; Ziegler et al., 2019; Stiennon et al., 2020).

## 3 Open Problems and Limitations of RLHF

Figure 1 (bottom) illustrates the categories of challenges and questions we cover in this section. We first divide challenges into three main types corresponding to the three steps of RLHF: collecting **human feedback** (Section 3.1), training the **reward model** (Section 3.2), and training the **policy** (Section 3.3). Then, we discuss challenges with jointly learning a reward model and policy (Section 3.4).

RLHF is not able (and was not designed) to be a solution by itself for developing safe AI. There are some problems that it is not equipped to solve, which will require other strategies (we discuss these in Section 4.3). To highlight this, we introduce a distinction between challenges with RLHF that are relatively **tractable** and could reasonably be addressed within the RLHF framework using improved methodology versus ones that

are more fundamental limitations of alignment with RLHF. **The key distinction between tractable and fundamental challenges is that fully overcoming fundamental ones is either impossible or would require a method that is no longer a form of RLHF.**[2] Although many of the fundamental problems we identify can be lessened by improving methodology, they could not be fully addressed with RLHF. As a result, they should be either avoided by not using RLHF or compensated for by other safety measures. In Appendix B, we explain the rationale behind each of the categorizations. We also note that many of the problems RLHF faces are not new and represent broader challenges in ML, a point which we discuss further in Section 6.

### 3.1 Challenges with Obtaining Human Feedback

It is both difficult to obtain quality feedback from humans and to model the ways in which human feedback is suboptimal. Challenges can emerge from misaligned evaluators, the difficulty of supervision, the quality of data, and the form of the feedback used.

#### 3.1.1 Misaligned Humans: Evaluators may Pursue the Wrong Goals

Humans can pursue harmful goals, either innocently or maliciously.

**Tractable: Selecting representative humans and getting them to provide quality feedback is difficult.** RLHF at scale requires selecting and instructing human evaluators. However, this has resulted in biases. Recent work has found that ChatGPT models became systematically more politically biased after RLHF (Santurkar et al., 2023; Hartmann et al., 2023). The exact cause of this bias remains unclear. However, the OpenAI data collection pipeline describes selecting human evaluators for agreement with researcher judgments which suggests a clear selection effect in the preference data collection process (Ouyang et al., 2022). Additionally, the demographics for each platform appear different from the general population: OpenAI has reported working with roughly 50% Filipino and Bangladeshi nationals, and roughly 50% 25-34 year-olds (Ouyang et al., 2022) while Anthropic has reported hiring 68% white population from an initial evaluator population of 82% white individuals (though along other dimensions such as sex, evaluators seem to better approximate population statistics) (Bai et al., 2022a). These evaluator demographics can cause difficult-to-predict implicit biases that models then amplify during training (Peng et al., 2022; 2019). Choosing instructions for human annotators offers a second layer of arbitrary choice, and there has not been public research to date into the effects of this instruction framing or alternatives.

**Tractable: Some evaluators have harmful biases and opinions.** Humans do not always have desirable and ethical opinions. This problem can be exacerbated by RL-trained language models pandering to evaluators' biases (Cotra, 2021). This is known as *sycophancy* (Perez et al., 2022b; Sharma et al., 2023), and it can worsen with model size (Amodei et al., 2016; Perez et al., 2022b). Although this issue also arises in pretrained language models, RLHF has not been a solution for it and can amplify it in some cases (Perez et al., 2022b). However, the extent to which it is caused by RLHF remains unclear.

**Tractable: Individual human evaluators can poison data.** Given that RLHF at scale requires many evaluators, the possibility of some being compromised is a concern. Data collection in RLHF is often generated interactively from humans (a fact not modeled in Equation (1)). This could be hazardous if an evaluator seeks to attack the model. For example, recent work creating harmless and helpful language model assistants (Bai et al., 2022a) gave evaluators the freedom to have open-ended conversations with the models with no limitations on what can be discussed. Rando and Tramèr (2023) show that this setup can be exploited by malicious annotators to create poisonous examples and inject an *universal jailbreak backdoor* that can elicit arbitrary undesired behaviors at inference time. A similar attack is successful for instruction tuning with very few examples (Wan et al., 2023; Xu et al., 2023a), and poisoning web-scale image datasets is possible under realistic assumptions (Carlini et al., 2023a).

---

[2]This distinction is soft, and some categories of challenges could arguably be either. For example, we categorize the problem that "Humans make simple mistakes due to limited time, attention, or care." (Section 3.1.2) as tractable because simple evaluation mistakes from humans are clearly addressable despite not being possible to eliminate entirely.

### 3.1.2 Good Oversight is Difficult

'Scalable oversight' refers to the ability to effectively supervise models given limited resources and bandwidth (Amodei et al., 2016). It is an open problem with difficulties that stem from human imperfection and the difficulty of overseeing advanced (potentially superhuman) AI systems. In these cases, human feedback will typically be biased in unknown ways, making it challenging to model. See also Bowman et al. (2022) which focuses in-depth on scalable oversight.

**Tractable: Humans make simple mistakes due to limited time, attention, or care.** Humans sometimes make mistakes due to factors such as lack of interest in the task, attention decay, time constraints, or human biases (Pandey et al., 2022; Chmielewski and Kucker, 2020; Hosking et al., 2023). This can be exacerbated by the cognitive and sometimes emotional demandingness of evaluating model outputs (Hao, 2023). Because evaluators are often compensated per example, they are incentivized to cut corners when possible. Mistakes can be correlated across annotators. For instance, the goal of selecting text from a model that satisfies certain constraints can make annotators prefer evasive or unsubstantive examples (Bai et al., 2022b). Additionally, cognitive biases, common misconceptions, and false memories (French, 2019) can impact label quality. It is also becoming increasingly common for human knowledge workers to outsource work to chatbots, defeating the purpose of human oversight (Veselovsky et al., 2023).

**Tractable: Partial observability limits human evaluators.** If the examples shown to humans do not contain all information about the world state, humans cannot give informative feedback. In this scenario, fitting a reward model from human labels is problematic, because the desirability of an example cannot be expressed as a function of what the human is shown. For example, Krakovna et al. (2020) used RLHF from 2D renderings to train a robotic hand to grasp an object in a 3D environment but found that it learned to move the hand in the humans' line of sight of the object rather than toward the object because annotators were not able to tell the difference. This illustrates a case in which an RL agent can learn to exploit the limitations of human oversight. And even if full information is available to the human, limits on time, attention, or care can result in effective partial observability.

**Fundamental: Humans cannot evaluate performance on difficult tasks well.** Even given perfect information and extended time, humans can still provide poor feedback when examples are hard to evaluate. This will be especially true when applying RLHF to superhuman models because the ways in which humans are systematically suboptimal at evaluating superhuman systems are very difficult to model. Saunders et al. (2022) find that human evaluators of a model trained to summarize passages miss over half of the critical errors and include substantial inaccuracies in the summaries the models produced despite having unlimited time to find such errors. Meanwhile, Perry et al. (2022) find that humans miss security vulnerabilities introduced by LLM code assistants. Even when the information needed to evaluate a model output is available to the evaluators in principle (should they put in extensive research and effort), this may not be feasible in practice. Bowman et al. (2022) formulate tasks on which nonexpert humans struggle to grade answers to questions accurately and argue that human feedback alone will not be sufficient to exercise scalable oversight for superhuman AI systems.

**Fundamental: Humans can be misled, so their evaluations can be gamed.** Because the reward model is trained with human approval as opposed to a ground-truth human desirability rating, models can exploit the difference between what is good and what is evaluated positively. Language models can imitate the persuasive and manipulative tactics of humans (Bai, 2023; Vincent, 2023; Griffin et al., 2023). In particular, language models trained with RLHF can sound confident even when they are incorrect (Snoswell and Burgess, 2022) which can lead humans to provide more positive feedback (Bowman et al., 2022; Hosking et al., 2023). These incentives to mislead also connect to broader worries about manipulation (Kenton et al., 2021; Carroll et al., 2023; Everitt et al., 2021). In addition to sounding confident, RLHF can contribute to sycophancy (Perez et al., 2022b; Sharma et al., 2023), or "gaslighting" of humans (Vincent, 2023). Misleading behavior will actively be incentivized by RLHF when humans can be tricked into mistakenly providing positive feedback (Carroll et al., 2022; 2023; Steinhardt, 2023).

### 3.1.3 Data Quality

Obtaining representative and helpful data is an open technical problem.

**Tractable: Data collection introduces biases.** Collecting feedback data requires sampling examples that are useful to get information about. Ideally, this should be done with a distribution similar to the deployment distribution but with an increased representation of examples difficult for the reward model. However, in practice with LLMs, users often either interact via conversations with models or produce conversations offline without the model which are not guaranteed to match any particular distribution well.

**Fundamental: There is an inherent cost/quality/quantity tradeoff when collecting human feedback.** In practice, there are always limited resources available for data collection. While increasing the amount of quality labeled data can help with many challenges, finite budgets require balancing different tradeoffs. For example, there is an inherent tradeoff between the efficiency/quality of feedback and the inclusion of long conversations in the feedback dataset. Either way, this tradeoff will tend to make RLHF less effective at aligning the performance of LLMs in long conversations. Helpful approaches for improving data quality have been to obtain samples that are diverse (Zhou et al., 2023), adversarial (Ziegler et al., 2022), and which the reward model is uncertain about (Christiano et al., 2017). However, active learning techniques in deep learning rely on heuristics for prediction confidence which can be unreliable (Gleave and Irving, 2022). Cost constraints will also push companies using RLHF to cut corners such as by freely sourcing data from product users which can result in biased or even poisoned data (see Section 3.1.1). Defining the notion of data diversity, understanding its relationship with data efficiency, developing effective methods for diverse data selection, and understanding how the amount of feedback data influences outcomes are all open problems.

### 3.1.4   Limitations of Feedback Types

**Fundamental: RLHF suffers from a tradeoff between the richness and efficiency of feedback types.** In addition to tradeoffs between cost, quality, and quantity that result from the overall process of feedback collection (see Section 3.1.3), similar tradeoffs result from the feedback modality. There exists an interplay between the choice of feedback type and what biases human labelers express (Bansal et al., 2023). These choices influence the final policy learned after RLHF, but it is unclear what feedback protocol to use for evaluating the resulting differences (Bansal et al., 2023). Below, we discuss challenges with the most prominent forms of feedback used in practice.

**Comparison-based feedback:** The most common type of feedback used with RLHF is binary preferences between pairs of examples (Christiano et al., 2017) though $k$-wise rankings (Brown et al., 2019; 2020; Zhu et al., 2023; Myers et al., 2021; Yuan et al., 2023) or best-of-$k$ queries (Biyik et al., 2019) can be used as well. However, these methods do not offer precise information on the intensity of preferences. A learned preference ordering can fail to converge to the true one when the desirability of examples depends on noise or unmodeled, contextual details not contained in the observations (e.g., randomness in a human's feedback or differences between evaluators (Myers et al., 2021)). Moreover, it is typically assumed that humans will select an option in proportion to the exponential of its utility (Christiano et al., 2017), but this is a convenient modeling assumption, not a principled one. Another problem is that comparison-based feedback will lead to policies that have a high median performance rather than a high average one. Consider a simple example in which actions of type $A$ are always recognized to be of value 1 to an evaluator, while actions type $B$ are recognized to have value 10 on 40% of examples but are overlooked and concluded to have value 0 on 60%. Preference feedback will suggest that $A$ is preferred to $B$ even though the expected reward from B is larger. See also Section 3.2.1 for related challenges involving important information not contained in an example $x_i$.

**Scalar feedback:** Obtaining scalar feedback addresses some problems of comparison-based feedback – it is significantly more expressive (Wilde et al., 2022). However, scalar rewards from humans can be poorly calibrated. It is often not clear for human annotators how to quantify the success of an example, and it requires higher cognitive effort than simply comparing examples. Scalar feedback is more susceptible to inconsistency between annotators and suffers from bias due to the order in which examples are presented (Yannakakis and Hallam, 2011). A combination of comparison and scalar feedback where the annotators indicated the intensity of a preference using a slider bar was demonstrated by Wilde et al. (2022), but it requires more sophisticated and annotator-specific human response models. Attempting to discretize this form of feedback using a Likert scale (a range of discrete ratings; e.g., very bad, bad, ok, good, very good) simplifies the process of feedback collection (Knox and Stone, 2008; MacGlashan et al., 2017; Arumugam

et al., 2019). However, the resulting learned preference ranking can be the opposite of the true one when assumptions commonly made in practice are violated (Ethayarajh and Jurafsky, 2022). Bansal et al. (2023) also discovered an inconsistency problem between scalar and ranking feedback in which the preferences inferred from ratings and rankings significantly disagree 60% for both human and AI annotators.

**Label feedback:** Sometimes, humans can provide feedback in the form of classifying examples. Label selection can be low-effort, but often suffers from *choice set misspecification* (Freedman et al., 2021; Guerdan et al., 2023; Casper et al., 2023b) when the given options don't fully encompass the labels needed to properly describe the data. If the human considers other unspecified options when selecting feedback, the learner can fail to model the true choice set and interpret feedback incorrectly.

**Correction feedback:** Feedback can come in the form of corrective demonstrations or adjustments that improve on an example from the model. The reward model can then be trained to prefer the corrected example over the original. In robotics, correction-based feedback has been used for improving policies (Li et al., 2021; Losey et al., 2022; Bajcsy et al., 2018) and plans (Sharma et al., 2022). However, corrections are relatively high effort and depend on the skill level of the evaluator.

**Language feedback:** Using language, humans can convey a large amount of information per evaluation, reducing ambiguity and goal misspecification. Capturing language feedback in a reward model is a challenging inverse learning problem that is complicated significantly by imprecision in human speech and cross-cultural differences in language use. A body of work on using language feedback for reward inference and shaping might lessen this challenge (Fu et al., 2019; Goyal et al., 2019; Sumers et al., 2021; Zhou and Small, 2021; Lin et al., 2022; Yu et al., 2023), but thus far, these techniques have not been applied to LLMs. The key drawback of approaches like these is that feedback is high-effort, reducing the number of examples that can be evaluated. This increases the risk of failing to spot rare failure modes. See also Section 4.2 for a discussion of related methods that use human language feedback for training LLM policies *without* using a reward model (which excludes them from our definition of RLHF).

## 3.2 Challenges with the Reward Model

Here, we discuss challenges resulting from misspecification, misgeneralization, reward hacking, and evaluating the reward model, $\hat{r}_\phi$. Each involves instances in which it can be difficult to train a good reward model, even from high-quality human feedback.

### 3.2.1 Problem Misspecification

The standard approach to fitting a reward model to represent human values is a doubly-misspecified problem.

**Fundamental: An individual human's values are difficult to represent with a reward function.** Unlike the model in Equation (1), human feedback can depend on contextual factors that cannot easily be accounted for in the examples $x_{i=1,...,n}$ used to train the reward model $\hat{r}_\phi$. Humans possess a range of intricate and context-dependent preferences that evolve over time and are difficult to model accurately. Models of human goals based on incorrect assumptions about human decision-making can impair reward inference (Hong et al., 2022). Even modeling human preferences with a reward at all, implicitly accepting the reward hypothesis (Silver et al., 2021), might be unwarranted (Skalse and Abate, 2022b; Bowling et al., 2023; Vamplew et al., 2022; Bobu et al., 2023). A number of studies have examined incorrect assumptions in various aspects of human models, such as their use of regret (Knox et al., 2022), the hypothesis space of reward models (Bobu et al., 2020; Biyik et al., 2020), and pedagogic behavior (Milli and Dragan, 2020). Skalse and Abate (2022a) formally study the effect of inverse reinforcement learning with a misspecified Boltzmann model, which is also common (Jeon et al., 2020). Most work in RLHF does not take into account personality and context-dependence of human preferences (Milano et al., 2021; Lindner and El-Assady, 2022), and Zhao et al. (2016) prove a mixture of reward functions cannot be identified from binary preferences without additional context. Different models for the human can also be better or worse for learnability (Knox et al., 2022). In particular, modeling human irrationalities can make reward learning difficult (Nguyen et al., 2017; Mindermann and Armstrong, 2018; Shah et al., 2019), leading to a trade-off between efficiency and accuracy. Finally, there are further challenges posed when feedback comes in different

modalities (e.g., demonstrations and preferences). Jeon et al. (2020) and Bıyık et al. (2022) propose ways of combining different types of information about human goals, but these approaches are sensitive to modeling assumptions about the human.

**Fundamental:** **A single reward function cannot represent a diverse society of humans.** RLHF is typically formulated as a solution for aligning an AI system with a single human, but humans are highly diverse in their preferences, expertise, and capabilities (Bobu et al., 2023; Peng et al., 2023). Evaluators often disagree: Stiennon et al. (2020), Ouyang et al. (2022), and Bai et al. (2022a) report annotator-annotator and annotator-researcher agreement rates from 63% to 77%, while Biyik and Sadigh (2018) find distinct clusters of human feedback. Attempting to condense feedback from a variety of humans into a single reward model without taking these differences into account is thus a fundamentally misspecified problem. Moreover, current techniques model differences among evaluators as noise rather than potentially important sources of disagreement (Baumler et al., 2023) (see Equation (1)). As a result, when preferences differ, the majority wins, potentially disadvantaging under-represented groups (Prabhakaran et al., 2021; Feffer et al., 2023; Kirk et al., 2023).

### 3.2.2 Reward Misgeneralization and Hacking

Reward models tend to be imperfect, and imperfection in reward models leads to reward hacking.

**Fundamental:** **Reward models can misgeneralize to be poor reward proxies, even from correctly-labeled training data.** There can exist many ways to fit the human feedback dataset $\mathcal{D} = \{(x, y)_{i=1,\ldots,n}\}$, even in the limit of infinite training data (Skalse et al., 2023). Reward models can compute reward using unexpected, possibly contingent features of the environment (Michaud et al., 2020) and are prone to causal confusion and poor out-of-distribution generalization (Tien et al., 2023). Reward learning algorithms can even produce reward models that fail to train new agents from scratch in various settings, raising concerns about their reliability as signals for policy learning (McKinney et al., 2023).

**Fundamental:** **Optimizing for an imperfect reward proxy leads to reward hacking.** Reward models can differ from humans due to misspecification (Section 3.2.1) and misgeneralization (Section 3.2.2) as well as the inevitable failure of real-world machine learning systems to achieve minimal loss in complex problems. Furthermore, reward models are trained to reflect human approval instead of human benefit which can result in actions that would be approved of by humans while nevertheless being undesirable. Applying strong optimization pressure for an imperfect proxy measure for a goal tends to cause poor performance on the underlying target goal (Hoskin, 1996; Manheim and Garrabrant, 2018; Gao et al., 2022). For example, without regularization penalizing the KL divergence between a base model and the finetuned model, LLMs undergoing RL often learn to output nonsensical text (Ziegler et al., 2019; Stiennon et al., 2020). This type of problem is known as "reward hacking", and has been observed in AI systems, including those trained with RLHF (Skalse et al., 2022; Krakovna et al., 2020). Skalse et al. (2022) show that unhackable proxies are very rare in complex environments, and Zhuang and Hadfield-Menell (2020) prove under mild conditions that reward hacking should be expected by default. Using a suite of environments Pan et al. (2022) find that reward hacking also becomes more likely as an agent's raw capabilities increase.

### 3.2.3 Evaluating Reward Models

**Tractable:** **Evaluating reward models is difficult and expensive.** When the true reward function is known, several methods can be used to judge the quality of the learned reward model (Gleave et al., 2020a; Wulfe et al., 2022). However, in most cases, reward modeling is used only when the true reward function is not known, making direct evaluation impossible. Hence, the reward model is typically evaluated in an *indirect* way by optimizing an RL policy using the learned reward model and then evaluating the generations from the RL policy. This makes the reward model evaluation intricately dependent on the policy optimization process which is inherently expensive and noisy. It is also not clear how robust a reward model evaluation is to many ad-hoc choices made in the policy optimization process: e.g., choice of RL algorithm, policy network architecture, compute spent, and other various hyperparameter choices (Gao et al., 2022). Another issue with indirect evaluation is that the evaluation signal for the reward model is the same as the training signal – human approval. As a result, training and evaluation failures will be correlated. Despite

the widespread use of indirect evaluation, it is not clear what choices in the policy optimization process are most influential for accurate evaluation of reward models.

### 3.3 Challenges with the Policy

Here, we discuss challenges from policy optimization, misgeneralization, power-seeking, and mode collapse. Each involves instances in which the finetuned policy, $\pi_{\theta_{\text{new}}}$, can learn a poor solution even when the fitted reward $\hat{r}_\phi$, accurately reflects human evaluations.

#### 3.3.1 Robust Reinforcement Learning is Difficult

Safety in deployment requires robust performance, yet it remains challenging simply to train AI systems using RL.

**Tractable: It is (still) challenging to optimize policies effectively.** RL agents must interact with the environment to collect their own data. This requires balancing exploratory and exploitatory behavior (Amin et al., 2021; Yang et al., 2021). Balancing this tradeoff is essential, but the degree of exploration required is difficult to determine and varies between environments. This is further complicated in settings with high-dimensional state/action spaces or sparse rewards (Ding and Dong, 2020). Balancing exploration and exploitation in deep RL remains a fundamental yet open challenge (Amin et al., 2021; Yang et al., 2021). Deep RL is unstable, and results are often highly sensitive to initialization and difficult to reproduce (Nikishin et al., 2018; Irpan, 2018; Henderson et al., 2018). This instability is attributed to multiple factors such as the random nature of exploration, the violation of the i.i.d assumption in data collection, the biased nature of value functions, and the general unpredictability of learning in deep neural networks (Amin et al., 2021). Uc-Cetina et al. (2023) overview methods and limitations for RL with LLMs in particular.

**Tractable: Policies tend to be adversarially exploitable.** Even when learned policies are trained with a perfect reward signal, perform well at the task they are trained for, and generalize to a wide range of scenarios, they can still perform poorly in adversarial situations. This is a pressing concern, as models deployed into the real world can be adversarially attacked by humans or other AI systems. Even "superhuman" policies can fail catastrophically against policies specifically designed to exploit them (Gleave et al., 2020b; Wu et al., 2021b; Wang et al., 2022). Adversarial policies can be found either by re-purposing existing deep-reinforcement learning algorithms or by manual human optimization in the case of prompt-injections and jailbreaks (Choi et al., 2022; Willison, 2023; Albert, 2023; Oneal, 2023; Li et al., 2023b; Wolf et al., 2023; Liu et al., 2023; Rao et al., 2023; Wei et al., 2023; Shen et al., 2023; Shah et al., 2023) for language-models. Black-box access to a model (e.g., via API access) is sufficient for many adversarial policy attack algorithms, though white-box access (enabled for example by open-sourced or leaked model weights) enables even stronger exploits (Kos and Song, 2017; Casper et al., 2022).

#### 3.3.2 Policy Misgeneralization

**Fundamental: Policies can perform poorly in deployment even if rewards seen during training were perfectly correct.** The deployment distribution can always differ from the training and evaluation distributions in real-world settings (Christiano, 2019). Even with a correct reward signal, a policy can learn to competently pursue the wrong goal whenever the true goal is correlated with other events. Shah et al. (2022); Di Langosco et al. (2022) and Hilton et al. (2020) study this type of failure in-depth. Shah et al. (2022) present an example scenario in which a systems trained with RLHF misgeneralizes to pursue the mechanism of reward administration itself instead of the intended goal.

**Fundamental: Optimal RL agents tend to seek power.** RL agents have an incentive to seek power when possible to help them accomplish their goals (Turner, 2021; Turner et al., 2019; Turner and Tadepalli, 2022; Ngo, 2022; Krakovna and Kramar, 2023; Ngo, 2022). Power-seeking (e.g. seeking more influence or compute) is typically framed as a concern with agentic AI systems that have more open-ended option spaces than a typical language model. However, versions of power-seeking can emerge from the way that RLHF is typically used to finetune LLMs. For example, a question-answering LLM trained with RLHF has the

incentive to learn to influence human interlocutors in order to avoid conversations about challenging topics. Sycophantic behavior from LLMs offers another example (Perez et al., 2022b; Sharma et al., 2023).

### 3.3.3 Distributional Challenges

There are challenges posed by the distribution of outputs produced by the model both before and after training.

**Tractable: The pretrained policy introduces biases into policy optimization.** RLHF in LLMs typically begins with a base model that has been pretrained on internet text. This base model is typically used both as the initialization for the RL policy network and the reference model for KL-regularization. Korbak et al. (2022b) formalizes how RL with these KL penalties can be viewed as a form of Bayesian inference with the base model determining the prior. While empirically useful, it causes the base model to significantly influence the final model. Using a base model that has been pretrained on web text is a convenient initialization – not a principled one. Moreover, internet text encodes harmful biases (e.g., about human demographics), which are then inherited by the downstream model (Weidinger et al., 2021). These biases can persist through RLHF training process. For example, if sounding confident and producing correct answers are correlated in the base model, the reward model will learn that sounding confident is good and reinforce this in the policy.

**Tractable: RL contributes to mode collapse.** RL finetuning decreases the diversity of samples produced by a model (Khalifa et al., 2021; Perez et al., 2022a; Glaese et al., 2022; Go et al., 2023) (a phenomenon known as "mode collapse"). OpenAI (2023) found that RLHF finetuning of GPT-4 harmed its calibration on question-answering. Santurkar et al. (2023) found LLMs finetuned with RLHF expressed a narrow distribution of political views. Mode collapse is plausibly due in part to switching from the supervised pretraining objective to an RL objective (Song et al., 2023). RL incentivizes the policy to output high-scoring completions with high probability, rather than with a probability in line with a training distribution. Addressing this is complicated because mode collapse can be beneficial or harmful in different cases. For example, it is desirable if an LLM assistant is 90% sure the answer to a question is "yes", it is better for the LLM to answer "probably" 100% of the time rather than answering "yes" 90% of the time and "no" 10% of the time. On the other hand, some preferences are inherently distributional (Khalifa et al., 2021; Weidinger et al., 2021) (e.g., gender balance). We discuss techniques that can be used to fine-tune LLMs with a distribution-matching objective in Section 4.2.3.

### 3.4 Challenges with Jointly Training the Reward Model and Policy

RLHF's dependence on training both a reward model and policy poses two unique problems.

**Tractable: Joint training induces distribution shifts.** Learning both a reward model and a policy is technically challenging – the reward model influences the learned policy, and the policy determines the distribution of the data used to train the reward. On one hand, if the reward model is trained on offline data, it is likely to misgeneralize (Levine et al., 2020). On the other hand, if reward and policy are learned jointly by gathering feedback from policy samples, the system will be prone to "auto-induced distributional shift" (Krueger et al., 2020; Li et al., 2023d). Features with overestimated rewards will become gradually more present in the feedback data, and features with underestimated rewards will disappear. This slows convergence during training. Thus errors from the reward model can also accumulate and become difficult to correct with feedback once the policy stops generating diverse alternatives (Wu et al., 2021a).

**Tractable: It is difficult to balance efficiency and avoiding overfitting by the policy.** The three key steps of RLHF can be performed synchronously, but in practice with LLMs, they are often performed serially. In this case, the reward model will typically be inaccurate off-distribution, which is precisely where the policy will learn to go (Gao et al., 2022; Levine et al., 2020). This is usually solved by obtaining fresh preference labels after a certain number of iterations of policy training. Appropriately setting this hyperparameter is important. Too low and information in the preference labels is wasted; too high and the policy navigates to unreliable regions of the reward model (McKinney et al., 2023; Christiano et al., 2017). Without a labeled validation set in the regions the policy is exploring, it is difficult to detect reward over-optimization during

training. Helpful approaches might include measuring KL-shift (Gao et al., 2022) or tracking the amount of disagreement in an ensemble of reward models.

# 4 Incorporating RLHF into a Broader Framework for Safer AI

Because of the challenges surveyed in Section 3, relying heavily on RLHF for developing safe AI poses risks. While RLHF is useful, it does not solve the fundamental challenges of developing human-aligned AI. More generally, no single strategy should be treated as a comprehensive solution. A better approach is defense in depth: multiple safety measures with uncorrelated failure modes. This is akin to assembling multiple layers of Swiss cheese—each has holes, but when layered can compensate for each other's failures (Hendrycks et al., 2021). While this type of approach is promising, it also comes with problems. For example, many of the challenges in Section 3 are not unique to RLHF, so it may be hard to find safety methods with uncorrelated failures. In this section, we discuss approaches that can be used to better *understand* (Section 4.1), *improve* on (Section 4.2), and *complement* (Section 4.3) RLHF in various ways as part of a broader agenda for AI safety.

## 4.1 Frameworks for Better Understanding RLHF

Although RLHF is becoming more widely used, there remain open questions about what factors are at play within it and how they influence the overall outcome. Here, we discuss approaches to address challenges for RLHF.

**Psychology and human-computer interaction.** Many of the open questions with RLHF involve the dynamics at play between humans and AI. It remains a challenge to understand the conditions which best allow for safe, reliable human-computer interaction. Specifically, it is unclear what type of feedback (or combination thereof) is best for learning human goals (Bansal et al., 2023), precisely how biases harm the quality of feedback, and how to best select and train human evaluators. As discussed in Section 3, human desires are difficult to express with a reward function (Skalse and Abate, 2022b; Bowling et al., 2023; Vamplew et al., 2022). Further work may be valuable toward inferring what beliefs humans are operating under and either asking for feedback while taking into account human uncertainty (Biyik et al., 2019) or correcting for human biases (Reddy et al., 2019; 2020; Chan et al., 2019; Tian et al., 2023). Reward modeling systems must also take advantage of techniques that distinguish between humans with different levels of expertise (Daniels-Koch and Freedman, 2022), confidence (Zhang et al., 2021), or noisiness (Freedman et al., 2023; Barnett et al., 2023).

**Sociology and social choice.** AI alignment must address not only individuals' perspectives, but also the norms, expectations, and values of affected groups. Some works have begun to assess whether LLMs can be used to facilitate agreement between different humans (Bakker et al., 2022) and to codify the broad-ranging principles under which deployment of AI systems for public good can be assessed (Floridi and Cowls, 2022; Sartori and Theodorou, 2022). The majority-rule problem with RLHF can also be improved by algorithms that explicitly model multiple evaluators (Freedman et al., 2023; Gordon et al., 2021; Davani et al., 2022; Daniels-Koch and Freedman, 2022; Gordon et al., 2022; Barnett et al., 2023), that tune models to individuals (Kumar et al., 2021), or that use more sophisticated aggregation strategies (Noothigattu et al., 2018). However, none of these approaches can solve the fundamental problem of how an AI system cannot be aligned to multiple groups of humans who hold conflicting viewpoints (Dobbe et al., 2021; Mishra, 2023). Many societies, however, confront this fundamental issue regularly. For example, democracies seek to reflect social preferences by soliciting the feedback of individuals. These systems generally fail to align diverse preferences yet tend to be more acceptable than less-democratic alternatives. As such, it is important to analyze RLHF from the lens of social choice theory (Sen, 1986) and work to understand whether the means by which it aggregates preferences is normatively acceptable.

**Assistance games.** Assistance games, such as cooperative inverse RL (CIRL) (Hadfield-Menell et al., 2016), provide a framework to analyze algorithms like RLHF. They offer a mathematical model to evaluate different design decisions in the communication of preferences to learning systems. In an assistance game, a human and an agent act together in the environment. Both seek to optimize the human's latent reward

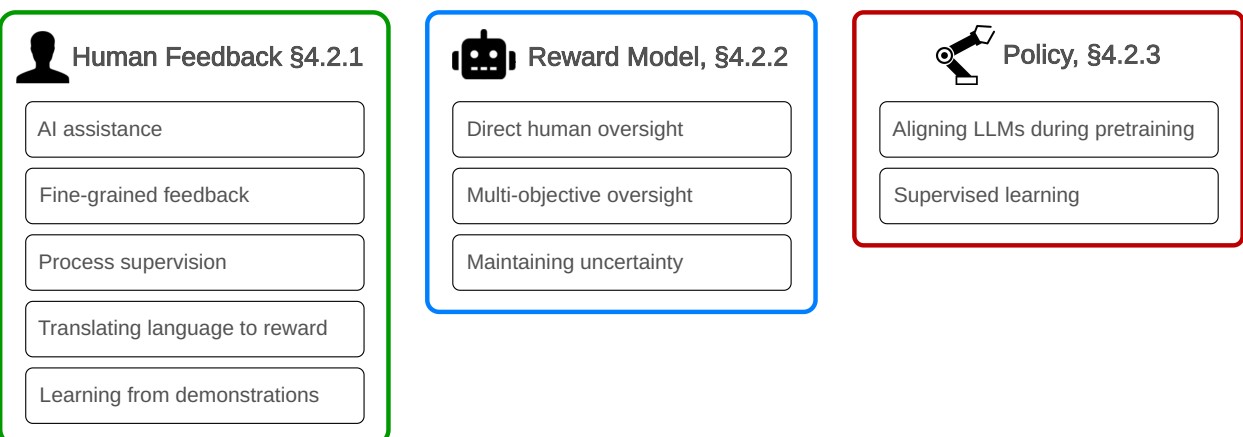

Figure 3: **Strategies that can be used to address various problems with RLHF.** Each approach is discussed in Section 4.2.

function, while only the human can directly query this reward function. In this model, querying the human is simply an additional action that the robot can take, and it is possible to study different querying strategies or profiles. Studying RLHF as an assistance game emphasizes the performance of the human-robot team. This might suggest alternative preference elicitation methods. Two examples are using active reward learning to determine when to collect feedback and which feedback to request first (Sadigh et al., 2017), and leveraging dialogue models to learn desired feedback-seeking patterns (Krasheninnikov et al., 2022). Of particular interest is understanding the consistency and convergence properties of RLHF, the impact of different error patterns from raters, and the effect of different rates of feedback.

**Bayesian inference.** Finetuning an LLM using RL with KL penalties on the differences between the pretrained model can be understood as a form of Bayesian inference: conditioning a prior (base LLM) on evidence about the desirable behavior of an LLM provided by the reward model (Korbak et al., 2022b). This perspective on RLHF separates the modeling problem (defining a target distribution specifying the desired behavior of an LLM) and the inference problem (approximating that target distribution) (Korbak et al., 2022a; Go et al., 2023). This can aid in answering questions about how the prior influences the outcome of RLHF. The typical target distribution of RLHF (a Boltzmann distribution) is a particular design choice and other distributions may address some of its limitations by, for example, differently fitting distributional preferences (Khalifa et al., 2021). Similarly, RLHF's inference algorithm (RL with KL penalties; equivalent to a variational inference approach (Korbak et al., 2022b)) could be replaced by a particular sampling strategy (e.g., rejection sampling or best-of-$n$ sampling).

**Worst-case behavior.** While RLHF seems to improve the average performance of a system, it is not clear what effects it has on worst-case behavior. It was not designed to make systems adversarially robust, and empirical vulnerabilities of systems trained with RLHF have been demonstrated with jailbreaks and prompt injection attacks (Choi et al., 2022; Willison, 2023; Albert, 2023; Oneal, 2023; Li et al., 2023b; Wolf et al., 2023; Liu et al., 2023; Rao et al., 2023; Wei et al., 2023; Shen et al., 2023; Shah et al., 2023). As a consequence, it would be valuable to better understand the worst-case behaviors of RLHF systems, potentially through the lenses of theoretical properties (Wolf et al., 2023; El-Mhamdi et al., 2022), decision theory (Casper, 2020), adversarial attacks (Perez et al., 2022a;b; Casper et al., 2023b; Ziegler et al., 2022; Carlini et al., 2023b), or rigorous evaluations (ARC, 2022; OpenAI, 2023; Shevlane et al., 2023).

## 4.2 Addressing Challenges with RLHF

Just as RLHF has challenges involving feedback (Section 3.1), the reward model (Section 3.2), and the policy (Section 3.3), there are various methods (both new and old) that can replace or supplement parts of the RLHF pipeline to address these types of challenges. Figure 3 outlines these methods. See also Wang et al. (2023) for a survey of methods for aligning LLMs.

### 4.2.1 Addressing Challenges with Human Feedback

**Providing feedback with AI assistance.** One way to amplify the abilities of humans is to have AI tools assist in generating feedback. Engineering prompts for an AI system and using it to automate feedback can substantially increase practicality and cost-effectiveness due to reduced reliance on humans. Nonetheless, AI-generated feedback still fundamentally depends on humans because (1) the models providing feedback are trained on human-generated data, and (2) humans control prompts and the process of incorporating feedback. There are several notable examples of AI-generated language feedback (Bai et al., 2022b; Saunders et al., 2022; Ye et al., 2023; Kim et al., 2023; Akyürek et al., 2023; Madaan et al., 2023; Chen et al., 2023; Gilardi et al., 2023; Lee et al., 2023) with research agendas like Recursive Reward Modeling (Leike et al., 2018) and AI Safety via debate (Irving et al., 2018; Du et al., 2023). However, AI-generated feedback has drawbacks. Humans often disagree with AI feedback. The rate of human/AI disagreement will vary by task, but Perez et al. (2022b), Casper et al. (2023b), and Lee et al. (2023) found this to happen up to 10%, 46%, and 22% of the time respectively in different experiments. Machines can also exhibit correlated failure modes not found in humans, such as vulnerabilities to some adversarial attacks. The extent to which AI feedback is a viable way to safely augment human feedback remains uncertain. However, it cannot theoretically be a comprehensive solution to AI alignment due to the bootstrapping problem behind ensuring the feedback-providing AI is aligned.

**Fine-grained feedback.** Many problems with feedback involve difficulty conveying precise information via the feedback signal (Section 3.1.4). To address this, Wu et al. (2023) and Cabi et al. (2019) use feedback on specific portions of examples and Wu et al. (2023) use feedback with respect to different goals of the model (e.g., correctness, relevance). This might improve the quality of the learned reward models at the cost of human feedback being more expensive to provide. Fine-grained feedback is not yet well studied nor widely adopted, so additional work to understand its advantages and feasibility will be valuable.

**Process-based supervision.** One challenge with training AI systems to solve problems is the difficulty of supervising performance on multi-step procedures. In RL, rewards can be very sparse for such problems. To address this, some works have trained LLMs to better solve multi-step math problems with process supervision (Uesato et al., 2022; Lightman et al., 2023). However, a fundamental challenge of both fine-grained and process-based supervision will be that the high-effort nature of the feedback trades off with the quantity that can be provided. This will make it more difficult to spot rare failure modes.

**Translating natural language specifications into a reward model.** Many issues with RLHF arise due to the difficulty of fitting a reward function using some constrained type of feedback. An alternative approach can be to generate a reward signal more directly from natural language directions, bypassing the need for feedback on examples. This approach could resemble a technique used by Bai et al. (2022b) which involved using prompts to guide an AI assistant to identify responses that violated certain user-defined specifications. Recently, Kang et al. (2023) demonstrate a zero-shot version of preference learning using LLMs, and Li et al. (2023a) introduced a method to elicit intended behavior through free-form, language-based interaction with users. Moreover, Luketina et al. (2019) surveys other possible techniques to accomplish this goal in non-LLM settings.

**Learning rewards from demonstrations.** An alternative approach to learning a reward model, known as inverse reinforcement learning (IRL) (Ng et al., 2000; Ramachandran and Amir, 2007; Ziebart et al., 2008), involves humans providing demonstrations instead of offering feedback on ones generated by the model. Jeon et al. (2020) and Bıyık et al. (2022) propose systematic ways of combining demonstrations, preferences, and possibly other types of human feedback to learn reward functions. While demonstrations carry rich information and avoid the need to have a system learn from its own generations, they are often more difficult

to gather because they require higher effort and expertise to perform the task. Additionally, the quality of demonstrations is limited by the talent of whatever expert is providing them, which warrants more research on learning from suboptimal human demonstrations (e.g., Brown et al. (2019); Zhang et al. (2021)).

### 4.2.2 Addressing Challenges with the Reward Model

**Using direct human oversight.** Although learning a reward model is efficient, it might be necessary to directly provide rewards (MacGlashan et al., 2017) for RL training in certain safety-critical situations.

**Multi-objective oversight.** Richer multi-objective signals that rate outputs on multiple objectives (Vam-plew et al., 2022) could lead to more flexible oversight. Current reward models assume that expert feedback is drawn from an underlying unimodal reward function (Barnett et al., 2023; Myers et al., 2021). But this is overly simplistic (Skalse and Abate, 2022b; Bowling et al., 2023). For instance, it can lead to a reward model that merely captures the preferences of the majority, and suppresses the preferences of minorities as noise. Using constraints (Malik et al., 2021; Lindner et al., 2023) or reward models that account for the diversity of preferences by assuming underlying reward functions to be multimodal (Myers et al., 2021; Bakker et al., 2022; Barnett et al., 2023; Siddique et al., 2023; Bhatia et al., 2020) can help mitigate this issue. Multi-objective oversight can also be useful for steering systems toward desired balances between competing values (e.g., helpfulness and harmlessness).

**Maintaining uncertainty over the learned reward function.** Given the challenges of accurately learning the appropriate reward function, several studies have emphasized the importance of taking uncertainty in the learned functions into account. Yue et al. (2023) and Liang et al. (2022b) tackle this by having the policy avoid types of states unseen by the reward model. Using an ensemble of reward functions has also been used to address these challenges (Christiano et al., 2017), demonstrating that this approach can enhance the diversity of text output (Rame et al., 2023) and its applicability for active learning (Gleave and Irving, 2022). Other strategies can include forms of risk-aversion (Hadfield-Menell et al., 2017) or handling uncertainty with a safe "shield" policy (Jansen et al., 2018; Srinivasan et al., 2020; Cohen and Hutter, 2020).

### 4.2.3 Addressing Challenges with the Policy

**Aligning LLMs during pretraining.** RLHF in LLMs typically begins by pretraining the LLM on internet text which includes a large amount of undesirable content. Korbak et al. (2023) argue that it can be more effective to use human feedback during pretraining by using a reward model to filter, weight, or annotate pretraining data. This also simplifies the process of aligning models by having them exhibit desirable behaviors from the outset rather than having them learn undesirable behavior and then attempt to unlearn it during finetuning.

**Aligning LLMs through supervised learning.** Several techniques for aligning LLMs with human preferences obtain results competitive with RLHF by using supervised learning to complement (Ramamurthy et al., 2022) or replace RL. The simplest variant of this is to perform standard supervised learning on well-curated data. Curation can involve filtering out bad demonstrations (Gehman et al., 2020; Welbl et al., 2021; Dong et al., 2023), compiling a small set of good demonstrations (Solaiman and Dennison, 2021; Sanh et al., 2022; Ibarz et al., 2018; Stiennon et al., 2020; Chung et al., 2022; Bıyık et al., 2022; Zhou et al., 2023), or generating good demonstrations using an LLM, e.g., after conditioning human feedback provided in natural language (Scheurer et al., 2022; 2023; Chen et al., 2023; Xu et al., 2023b). A different family of methods augments the language modeling objective to utilize feedback provided by the reward model (Korbak et al., 2023; Yuan et al., 2023; Rafailov et al., 2023). This last setting shares similarities with offline RL, which focuses on training an optimal policy using demonstrations annotated with rewards (Levine et al., 2020; Snell et al., 2022; Hu et al., 2023).

### 4.3 RLHF is Not All You Need: Complementary Strategies for Safety

Other technical approaches to AI safety should be studied and implemented alongside RLHF. Establishing trust with AI systems should be approached with a combination of principled design choices, rigorous testing, interpretability, verification, and theoretical guarantees where possible (Leike et al., 2018). See also Critch

and Krueger (2020), Hubinger (2020), Hendrycks et al. (2021), and Ngo (2022) for additional overviews of strategies for building safer AI.

**Robustness.** As discussed in Section 3.3, models trained with RLHF can still exhibit undesired behavior due to distributional shifts between training and deployment. For example, adversarially engineered user inputs cause an LLM to output harmful text. To mitigate this problem, developers should use tools to generate inputs which result in undesired behavior and train against these adversarial examples (Zhang and Li, 2019; Ziegler et al., 2022; Perez et al., 2022a; Casper et al., 2023b). Anomaly detection techniques (Omar et al., 2013) can also be useful for flagging abnormal inputs likely to trigger bad behavior. Ensuring the security of important AI training runs against malicious human evaluators and/or outside cybersecurity threats will also be valuable.

**Risk assessment and auditing.** Although training processes should be crafted to produce models that are safe by design, evaluations are another layer of defense. Passing an evaluation is not proof of safety, but as is the case in almost every safety-critical industry, rigorous evaluations of capabilities and risks helps to spot hazards and establish trust. In practice, this should involve both in-house and second-party evaluations (OpenAI, 2023; ARC, 2022; Perez et al., 2022b). As with adversarial training for robustness, the development of improved red teaming techniques will be important (Perez et al., 2022a; Casper et al., 2023b).

**Interpretability and model editing.** Generating human-understandable explanations for the behavior of AI systems is currently an unsolved problem. Progress in explainability and interpretability could help verify hypotheses about how models make decisions (Geiger et al., 2023), including whether the decision-making process is trustworthy. In this way, it could be possible to gain confidence that models will (or will not) behave in a safe way without necessarily conducting extensive testing of the models (Jacovi et al., 2021). Red-teaming can also be complemented by interpretability techniques (Rastogi et al., 2023; Räuker et al., 2023), especially for purposes of identifying adversarial inputs (Ziegler et al., 2022; Casper et al., 2023c;a) or anomalous inputs (Pang et al., 2021). In another direction, better understanding the internal mechanisms of models can aid in directly editing model weights or intervening on internal activations in order to improve truthfulness (Li et al., 2023c), modify a model's factual knowledge (Meng et al., 2023; 2022; Hernandez et al., 2023; Hase et al., 2023), or otherwise steer model behavior (Cui et al., 2022).

## 5 Governance and Transparency

Social scientists and policymakers have increasingly focused on the need for governance frameworks to develop and deploy AI systems responsibly. Across historical contexts, a hallmark of mature scientific fields is the open sharing of research findings (Shapin and Schaffer, 2011) to allow experts to understand progress (Gilbert and Loveridge, 2021). Below we overview components of an RLHF governance agenda, including outstanding questions and risk dimensions.

**Incentives and requirements for safety.** Competition between labs can generate harmful race dynamics (Dafoe, 2018) because of tradeoffs between competitiveness and caution. This suggests a role for governance in promoting a healthier environment for safe AI research, development, and deployment (Dafoe, 2018; Perry and Uuk, 2019; Falco et al., 2021; Cihon, 2019; Anderljung et al., 2023). Governance in this form could involve mandates for independent auditing, evaluations, and certification (Shavit, 2023; Mökander et al., 2023; ARC, 2022; Hadfield and Clark, 2023; Shevlane et al., 2023); monitoring for post-deployment problems (Hendrycks and Gimpel, 2016); influence over resources including hardware and data (Brief, 2020; Chan et al., 2023a); and prohibiting deployment unless critical standards are met, as in the case of the U.S. Food and Drug Administration's oversight of clinical trials for testing potential new treatments (Junod, 2008).

**Transparency and auditing.** A sustained commitment to transparency would make the existing RLHF research environment more robust from a safety standpoint. First, the disclosure of some details behind large RLHF training runs would clarify a given organization's norms for model scrutiny and safety checks. Second, increased transparency about known efforts to mitigate risks could improve safety incentives and suggest methods for external stakeholders to hold companies accountable. Third, and most relevant for the present paper, transparency would improve the AI safety community's understanding of RLHF and

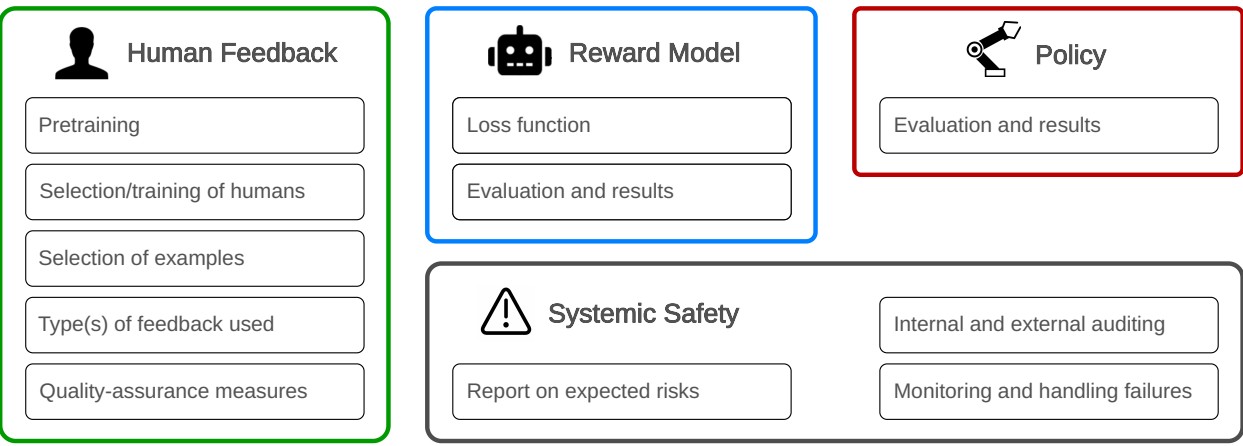

Figure 4: **Details behind an implementation of RLHF that, if disclosed, could be indicative of risks.** See Section 5 for a complete discussion. Companies using RLHF to train models for high-stakes or safety-critical applications should maintain transparency with the public and/or auditors about key details of their approach.

support the ability to track technical progress on its challenges (Clark, 2022). Some level of disclosure is a precondition to evaluate the viability of the technical RLHF safety agenda over time and allow for community contribution to it. For all of these reasons, working to incorporate transparency standards into an AI governance framework will be important (Larsson and Heintz, 2020; Anderljung et al., 2023).

It is possible that public disclosure of details critical to the development of model capabilities might lead to the unwanted proliferation of AI technologies that could be misused. However, detailing safety measures will often not require divulging implementable details, and when it does, private disclosure to second-party auditors (Mökander et al., 2023; ARC, 2022; Hadfield and Clark, 2023; Shevlane et al., 2023) offers a solution.

As more specific policy prescriptions are beyond our scope, we encourage elaboration on these topics as part of a future research agenda. Below, however, we outline specific types of details that, if disclosed, could be indicative of risks and should be accounted for when auditing AI systems developed using RLHF. See also Figure 4.

**Human feedback** details:

- **A description of the pretraining process including details about what data was used** to make apparent possible biases that pretraining can cause.

- **How human evaluators were selected and trained** to provide information about risks of evaluators being malicious, unrepresentative, or incapable.

- **The process by which examples were selected to obtain feedback** to invite scrutiny about their representativeness and whether sufficient adversarial training was used. If examples were crowdsourced from a publicly-available application, details about what measures were taken to avoid data poisoning attacks should be provided.

- **The type(s) of human feedback used** (e.g., binary comparisons, scalar feedback, etc.) to suggest what risks might be caused by insufficiently abundant or rich feedback.

- **A report on measures taken for quality assurance in feedback collection and inter-rater consistency** to ensure that effective quality control measures were taken.

**Reward model** details:

- **The loss function used to fit the reward model and how disagreement was modeled** (e.g., as noise) to help with analyzing the degree of misspecification when fitting the reward model.

- **A report on reward model evaluation and results** to suggest possible problems from a misaligned reward model. The evaluation should involve red teaming.

**Policy** details:

- **A report on policy evaluation and results** to suggest possible troubles from a misaligned policy. The evaluation should involve red teaming and include assessment for risky capabilities (e.g., the ability to deceive a human).

Systemic safety measures

- **A report on internal and external audits and red teaming** to ensure accountability and disclose risks that are identified.

- **A report on expected risks and anticipated failure modes** to ensure accountability.

- **Plans for monitoring and correcting failures that emerge** to support post-deployment safety.

How these types of risks should be documented remains an area of active work in AI governance. Similar questions have been asked in an investigation by the US Federal Trade Commission into OpenAI (FTC, 2023) but in response to problems with ChatGPT rather than proactively. Salient documentation proposals focus on regular reporting of reward components (Gilbert et al., 2022) and the ability to compare the capabilities of language models according to standard benchmarks (Liang et al., 2022a). For the longer term, incorporating beneficial standards for safety and transparency into norms and regulations affecting AI is an ongoing challenge.

**Concerns for social and economic equity.** Although this paper has focused on technical challenges with RLHF, there are social and economic ones as well which governance and industry should work to address. For example, OpenAI has paid Kenyan knowledge workers at a rate of less than $2 USD per hour (Perrigo, 2023) for work which was mentally and emotionally distressing and had negative impacts on their mental wellbeing (Hao, 2023). Human subjects used in RLHF research should not be systematically selected simply for their availability or low cost (National Commission for the Protection of Human Subjects, 1978). Costs, benefits, and influence over RLHF models should be equitably distributed across different communities (Whittlestone et al., 2021; Eloundou et al., 2023). There is an additional possibility that powerful AI systems will be highly profitable and serve to concentrate large amounts of wealth and power into the hands of a few (O'Keefe et al., 2020; Chan et al., 2023b). Thus, policies that address inequalities and protect vulnerable populations (e.g. impacted communities, whistleblowers) will be increasingly important.

## 6  Discussion

**While some problems with RLHF are tractable, others are fundamental and cannot be fully solved.** Technical progress in some respects is tractable, and this room for progress should be seen as a cause for concerted work and optimism. Even some of the fundamental problems that we overview can be alleviated with improved methodology even though they cannot be fully solved by RLHF. However, the fundamental nature of these problems requires that they be avoided or compensated for with non-RLHF approaches. Hence, we emphasize the importance of two strategies: (1) evaluating technical progress in light of the fundamental limitations of RLHF and other methods, and (2) addressing the sociotechnical challenges of aligning to human values by committing to both defense-in-depth safety measures and openly sharing research findings with the wider scientific community.

**RLHF = Rehashing Lessons from Historical Failures?** RLHF offers new capabilities but faces many old problems. The individual components of it (preference elicitation, fitting a reward model, and policy optimization) have a history of technical and fundamental challenges in the fields of human-computer interaction

and AI safety (e.g. (Lewis, 1990; Sutton and Barto, 1999; Ng et al., 1999)). In 2023, RLHF was described by the first author of Christiano et al. (2017) as a "basic solution" intended to make it easier to "productively work on more challenging alignment problems" (Christiano, 2023).[3] Some challenges and questions that we have covered are rather unique to RLHF such as ones involving jointly training the reward model and policy (Section 3.4). However, many other problems are instances of broader ones in machine learning such as challenges with RL policies (Section 3.3). Others still are fundamental problems with AI alignment such as determining whose values are encoded into AI in a diverse society of humans (Section 3.2.1). The successes of RLHF should not obfuscate its limitations or gaps between the framework under which it is studied and real-world applications (see Appendix A). An approach to AI alignment that relies on RLHF without additional techniques for safety risks doubling-down on flawed approaches to AI alignment. Thus, it will be important to continue working to better understand RLHF while respecting its limitations.

**Moving forward.** RLHF has clear advantages for aligning AI systems with human goals. As a result, it has been key to the development of state-of-the-art LLMs and will likely continue to play a major role in modern AI. However, its use and influence should be accompanied by a commensurate research effort to better understand RLHF and address its flaws. Because it optimizes for human approval, RLHF in particular demands a special type of caution because many of its failures will actively tend to be ones that humans struggle to notice. It will be important to approach RLHF cautiously and work to incorporate it into a more holistic framework (Khlaaf, 2023) for safer AI with multiple layers of protection from failures (Hendrycks et al., 2021). Because some of the challenges with RLHF are fundamental to the AI alignment problem itself, moving forward will require confronting the basic choices and assumptions behind any given approach to aligning AI and who controls it (Dobbe et al., 2021). Moving forward, we urge that those working to develop advanced LLMs using RLHF both contribute toward resolving its open challenges and maintain transparency about the details of their approach to safety and any anticipated risks.

## Contributions

Stephen Casper and Xander Davies served as the central writers and organizers.

Claudia Shi, Thomas Krendl Gilbert, Jérémy Scheurer, Javier Rando, Rachel Freedman, Tomasz Korbak, David Lindner, Pedro Freire, Tony Wang, Samuel Marks, Charbel-Raphaël Segerie, Micah Carroll, Andi Peng, Phillip Christoffersen, Mehul Damani, Stewart Slocum, Usman Anwar, Anand Siththaranjan, Max Nadeau, Eric J. Michaud, Jacob Pfau, Xin Chen, Dmitrii Krasheninnikov, Lauro Langosco, and Peter Hase contributed to planing and writing the paper.

Erdem Bıyık, Anca Dragan, David Krueger, Dorsa Sadigh, and Dylan Hadfield-Menell served as advisors.

## Acknowledgements

We thank Sam Bowman, Adam Jermyn, Ethan Perez, Alan Chan, Gabriel Recchia, Robert Kirk, and Nathan Lambert for their helpful feedback. This work was facilitated in part by the Harvard AI Safety Team and MIT AI Alignment group.

---

[3]Christiano (2023) mentions debate (Irving et al., 2018) and recursive reward modeling (Leike et al., 2018) as examples of 'more challenging alignment problems.' See also an outline of proposals in Hubinger (2020).

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

# A  An Improved Model of the Human Feedback Process

As illustrated in Equation (1), the feedback process in RLHF is typically modeled with a single human $\mathcal{H}$ with internal reward function $r_\mathcal{H}$; examples sampled from the base model: $x_i \sim \pi_\theta$; and feedback as a function of the human, example, and noise: $y_i = f(h, x_i, \epsilon_i)$. However, as discussed in Section 3, this is a misspecified model of the process: there is not a single human, humans values are not representable with a reward function, human actions are dependent on context, and the sampling process can involve a human. Thus we propose an alternative formulation.

Let $\Delta\mathcal{H}$ refer to a joint distribution of humans (or groups thereof if feedback is provided collaboratively) used for obtaining samples and feedback denoted as $\mathcal{H}_j^{\text{sample}}$ and $\mathcal{H}_j^{\text{feedback}}$. A dataset of examples is sampled from $\pi_\theta$ (or some other source) where each example $x_i$ is defined to be a batch of one or more generations from the base model. Importantly, $x_i$ may not contain all information about the world state (e.g., if $x_i$ is a 2D rendering of a 3D environment), and the human may be able to observe more than just the model's output (e.g., if interpretability tools are used to aid in evaluation). So let $v$ be a rendering function that maps $\pi_\theta$ and $x_i$ to what a human sees. The behavior of humans varies over time and in different contexts, so let $c_i^{\text{sample}}$ and $c_i^{\text{feedback}}$ represent particular contexts for sampling and feedback collection. Denote the sampling process as $s$ which maps the base model $\pi_\theta$, a human $\mathcal{H}_j^{\text{sample}}$, and context $c_i^{\text{sample}}$ to some example $x_i$. Notably, $s$ could ignore the base model and generate offline samples from some other source. Finally, let $f$ map a human $\mathcal{H}_j^{\text{feedback}}$, rendered example $v(\pi_\theta, x_i)$, and context $c_i^{\text{feedback}}$ to feedback $y_i$. The data collection process can thus be more completely modeled as:

$$\mathcal{H}_j^{\text{sample}}, \mathcal{H}_j^{\text{feedback}} \sim \Delta\mathcal{H}, \qquad x_i \sim s(\pi_\theta, \mathcal{H}_j^{\text{sample}}, c_i^{\text{sample}}), \qquad y_i = f(v(\pi_\theta, x_i), \mathcal{H}_j^{\text{feedback}}, c_i^{\text{feedback}}) \qquad (4)$$

which highlights a need for future work to better account for the aspects of this process that are commonly not accounted for when training systems with RLHF.

# B  Explanation for Tractable/Fundamental Classifications

In Section 3, we categorize problems as **tractable** or **fundamental**. The key distinction between the two is that fundamental challenges are substantial enough that overcoming them would require a method that is no longer a form of RLHF. Although many of the fundamental problems we identify can be alleviated by improving how RLHF is approached, they could be fully addressed with RLHF. As a result, they should be either avoided by not using RLHF or compensated for by other safety measures. This distinction is soft, and some categories of challenges are marginal. Here, we briefly explain each categorization.

**Problems from Section 3.1:**

**Tractable: Selecting representative humans and getting them to provide quality feedback is difficult:** This can be addressed by studying and improving the selection and training of evaluators.

**Tractable: Some evaluators have harmful biases and opinions:** This can be addressed by studying and improving the selection and training of evaluators.

**Tractable: Individual human evaluators can poison data:** This can be addressed with improved evaluator selection and quality assurance measures.

**Tractable: Humans make simple mistakes due to limited time, attention, or care:** This is marginal because human mistakes can never fully be overcome. However, they can be addressed with improved working conditions and quality assurance procedures.

**Tractable: Partial observability limits human evaluators:** Human evaluators can be provided with all information available in the policy's observations (although representing this in an easily-comprehensible way may be challenging).

**Fundamental: Humans cannot evaluate performance on difficult tasks well:** Human intelligence and cognitive capacity are limited. Humans cannot be expected to properly evaluate the performance of superhuman models on complex tasks. Thus, solving this problem would require no longer using human feedback in the way that RLHF does.

**Fundamental: Humans can be misled, so their evaluations can be gamed:** Human fallibility cannot fully be overcome, especially against optimization pressure from the learned policy.

**Tractable: Data collection can introduce harmful biases:** This can be addressed with improved data curation.

**Fundamental: There is an inherent cost/quality tradeoff when collecting human feedback:** This tradeoff is unavoidable in practice – obtaining diverse and high-quality examples (e.g. from long chatbot conversations) requires more effort.

**Fundamental: RLHF suffers from a tradeoff between the richness and efficiency of feedback types:** This tradeoff is unavoidable for data collection in practice – richer annotations require more effort.

**Problems from Section 3.2:**

**Fundamental: An individual human's values are difficult to represent with a reward function:** This problem is marginal. It can be improved in practice by improved modeling, but RLHF-based solutions will be limited by the intractability of perfectly modeling context and troubles with the reward hypothesis (Skalse and Abate, 2022b; Bowling et al., 2023).

**Fundamental: A single reward function cannot represent a diverse society of humans:** Trivial. Instead of being a fundamental limitation with RLHF, this is a broader limitation of AI alignment itself.

**Fundamental: Reward models can misgeneralize to be poor reward proxies, even from correctly-labeled training data:** This problem is marginal because it can and should be addressed by improved sampling in practice. However, it is impossible to perfectly represent a distribution with infinite support from a finite sample. Additionally, the deployment distribution will always differ from the training and evaluation distributions in real-world settings (Christiano, 2019).

**Fundamental: Optimizing for an imperfect reward proxy leads to reward hacking:** If a reward model is imperfect, reward hacking will always be a possibility from RL.

**Tractable: Evaluating reward models is difficult and expensive:** This can be addressed by performing thorough and expensive evaluations.

**Problems from Section 3.3:**

**Tractable: It is (still) challenging to optimize policies effectively:** This can be addressed with advancements in RL methodology.

**Tractable: Policies tend to be adversarially exploitable:** This problem is marginal because achieving certified adversarial robustness against practical threat models has empirically been intractable. Nonetheless, this can be addressed with robust optimization techniques.

**Fundamental: Policies can perform poorly in deployment even if rewards seen during training were perfectly correct:** This problem is marginal because it can and should be addressed by improved sampling in practice. However, it is impossible to perfectly represent a distribution with infinite support from a finite sample. Additionally, the deployment distribution will always differ from the training and evaluation distributions in real-world settings Christiano (2019).

**Fundamental: Optimal RL agents tend to seek power:** Power is instrumentally useful for agents.

**Tractable: The pretrained policy introduces biases into policy optimization:** This can be addressed with improved base models.

**Tractable: RL contributes to mode collapse:** This can be addressed with forms of RL that optimize for distribution-matching in desired instances.

**Problems from Section 3.4:**

**Tractable: Joint training induces distribution shifts:** This can be mitigated with synchronous learning or other strategies.

**Tractable: It is difficult to balance efficiency and avoiding overfitting by the policy:** This can be addressed with improved training methodology.

