# OpenReview forum: "Open Problems and Fundamental Limitations of Reinforcement Learning from Human Feedback"
_TMLR — Accepted by TMLR_

### Review · Reviewer_Y1DJ · 2023-10-09

**Summary Of Contributions:**

This paper is a rather large-scale review of the existing literature on Reinforcement Learning from Human Feedback, applied to fine-tuning large language models (LLMs). The paper is divided in two main sections:

- An identification of the problems in RLHF for finetuning LLMs (with, at first, some explanation as to why Reinforcement Learning is used for that). The problems being identified are thoroughly grounded in the existing literature and recent experiments. The problems are grouped by category.
- The proposition of some solutions or areas of research to alleviate the problems identified in the first part of the paper. Here again, this is not a simple discussion, but a thorough review of what other papers present as possible solutions, results of experiments, and existing discussions.

**Audience:**

Yes

**Broader Impact Concerns:**

The broader impact of this work is overall positive, because the point of the paper is to discuss biases and limitations in how humans are used to fine-tune LLMs.

**Claims And Evidence:**

Yes

**Requested Changes:**

My only requested change is addressing the weakness mentioned above, the fact that the background focuses too much on mathematical notations and general concepts ($\pi_{\theta}$ that generates samples, but what are samples? How does it work in practice? How does it integrate with the neural architecture of the LLM). I would suggest taking one LLM, for instance LLama (so that it is not always ChatGPT), and summarizing how it was pre-trained, then how it was fine-tuned.

**Strengths And Weaknesses:**

The paper is at first a bit surprising because it does not contain a technical contribution. However, the paper is generally very well-written, and the amount of work done to collect all the references and discuss that is commendable! Overall, I think that this paper provides significant information to people who fine-tune LLMs, and policy-makers around LLMs.

Strengths:

- The references discussed by the paper are numerous and diverse
- The structure of the paper is very clear
- The paper is very well-written and easy to read. It is a clear tool to understand the current status of fine-tuning LLMs

Weaknesses:

- The background section lacks intuition. It is very mathematical even though the rest of the paper seems tailored to a broader audience (not just maths and technical people). The background section lacks simple sentences that explain what LLMs are, what they are useful for, where they are used (existing products), where they already failed (examples of products misbehaving for instance), where RL is used in all that, and how it is done. The last two points are extensively presented using formulas, but the intuition is difficult to get from the formulas.

---

> ### Author Response · Authors · 2023-10-27
> **Reply to Y1DJ**
>
> Thank you! We appreciate the feedback, and we are glad that you found the paper well-written and think that it will offer useful information for practitioners and policymakers.
>
> We have posted an update to the paper and describe new changes to it in our reply to you and the other two reviewers.
>
> Thank you for pointing out possible improvements to the presentation in the background section. We did not remove any of the notation but opted to add a new subsection titled “High-Level Overview” with a nontechnical description of the process.
>
> - In the new subsection, we discuss how RLHF can be used for many types of tasks, but the application we focus on the most is chat-based assistants such as GPT4, Claude2, Bard, and Llama2.
> - Also in the new subsection, we added a plain-English description of (1) RLHF in general and (2) RLHF for LLMs with binary-preference feedback.
> - We also made corresponding updates to the captions of figures 1 and 2 to fully describe the processes depicted.
> - In the formal descriptions, we now also describe the processes of supervised pretraining, collecting feedback, fitting the reward model, and optimizing the policy in more lay terms. For each, this now includes a high-level description of the central goal of each step.
>
> Please let us know if you recommend other work on the paper. Thanks!

---

### Review · Reviewer_XNWD · 2023-10-20

**Summary Of Contributions:**

The authors propose an overview of Reinforcement Learning from Human Feedback (RLHF) as used to fine-tune Large Language Models (LLM). The submission lays out three clearly defined contributions:
1/ a survey of open problems that RLHF is currently facing. This survey is broken down into the three main RLHF components: data collection, reward modelling, and policy training.
2/ an overview of techniques to understand, improve, and complement RLHF state of the art,
3/ a proposition for auditing and disclosure standards to improve societal oversight of RLHF-powered LLMs (arguably this section relates more to LLMs than to RLHF).

**Audience:**

Yes

**Broader Impact Concerns:**

The whole document is somehow motivated by Broader Impact Concerns.

**Claims And Evidence:**

Yes

**Requested Changes:**

Please address the major and minor points I raised.

**Strengths And Weaknesses:**

I will start by saying that this overview of RLHF in LLM is necessary as all this is pretty new and hyped and it is sometimes difficult to navigate the literature. And the submission does a good job at it in general. But this does not prevent it from having several flaws in my opinion, which I enumerate below categorized by severity.

Major:
- It is a paper on RLHF for LLMs. This needs to be clear in the title. Most criticism is justified by the expectations we have on the end LLM product. Similarly, some sections seem more dedicated to LLMs than RLHF. I think that the paper needs to do a better job at identifying what is being discussed.
- Several omissions in the historical literature of the domain. First, RLHF existed before Christiano2017 at least in the dialogue literature, where evaluation was an identified bottleneck for ML and RL for a long time []. It connects to several domains such as human-in-the-loop RL [], dialogue iteration score [], preference-based RL [], inverse RL [], reward shaping, etc. Also, it needs to be more clearly stated that RLHF is actually an Offline RL setting and that overfitting/overoptimizing is a well-known problem that can be addressed for instance with uncertainty estimates [].
- I disagree with many Tractable/Fundamental flags. I would recommend to remove them because they are very subjective, don't bring much to the table, and yield confusion (even though I know that the choice is motivated in the Appendix).
- 4.2: Many of the way to address the challenges with RLHF seem to be going backwards. For instance with "Learning rewards from demonstrations", we specifically use RLHF because the use of demonstration is inefficient. Something hybrid could be tried: handmade corrections of the system generation, but the authors present the approach as an "alternative".

Minor:
- 3.1.3 "Data collection can introduce harmful biases": it is motivated that there is a distributional shift, but the connection to harmful biases is not established.
- 3.1.3 "There is an inherent cost/quality tradeoff when collecting human feedback": I would call it "cost/quality/quantity tradeoff" and it is unclear how it is a problem for RLHF. It means that more RLHF implies better performance.
- 3.1.4: I would argue that this belongs more to the challenges with the reward model since the choice of type of feedback is more limited by the ability to train a good reward model on top of it.
- 3.2.1 "A single reward function cannot represent a diverse society of humans": The problem is more on the data collection side imho. The problem is not that much that a single reward function cannot represent a diverse society of humans since the reward model could adapt to a specific user profile, the problem is that the annotator cannot annotate in accordance to a user profile, instead the annotator may consider their own user profile as an annotating guidance.
- 3.2.2 "Reward models can misgeneralize to be poor reward proxies, even from correctly-labeled training data": This is ML. How is that specific to reward modelling?
- 3.3.1 "It is (still) challenging to optimize policies effectively": this is more about training Offline RL policies, which is one order less effective than RL policies.
- 3.3.2 "Optimal RL agents tend to seek power": I did not understand this point.
- 3.3.3 "The pretrained model introduces biases into policy optimization". The last two lines are a very good example, but it is a problem of reward model generalization, how is that related to this specific problem?
- 3.3.3 col2 last paragraph: "Mode collapse is plausibly due in part to switching from the supervised pretraining objective to an RL objective" => this is because we move from a distributional objective to an objective function maximization. We could provide the RL agent with this kind of objective (and this is kind-of already the case as the policy is regularized towards the pretrained one).
- 3.4 "Joint training induces distribution shifts": "and features with underestimated rewards will disappear" => this is an exploration problem. This is the problem of online RL in real world scenarios: we don't want to let it explore low rewards actions, which includes the undersestimated ones. I don't see how it relates to its open problem.

Typos/language:
- p1 col1 2nd line from bottom: "down) (Perez" =>"down, Perez".
- p5 col1 1st line of 3.1.1: "innocently" => "unintentionally" (other propositions could be inadvertently, candidly, accidentally, unknowingly, mistakenly, carelessly)
- p9 col1 last paragraph: the first half is very repetitive and redundant.

---

> ### Author Response · Authors · 2023-10-27
> **Response to XNWD + updates**
>
> Thank you! We appreciate the feedback, and we are glad that you found the paper to do a good job at overviewing the space, even though RLHF is a rapidly developing topic.
>
> We have posted an update to the paper and describe new changes to it in our reply to you and the other two reviewers.
>
> Replies to major items in order
>
> 1. Re: Much focus on LLMs – Because LLMs are the most prominent application of RLHF today, it is somewhat challenging (and not necessarily desirable) to separate the study of challenges with RLHF and today’s LLMs. However, (1) all of the bold-headed paragraphs in section 3 about problems with RLHF apply to non-LLM RLHF, and (2) **we added an additional mention to the intro that the reason we focus on LLMs is because of current SOTA applications.**
> 2.
> - Re: Background on RLHF – We agree about the importance of not beginning the story of RLHF with Christiano et al. (2017). In the third paragraph of the introduction, we described Christiano et al. (2017) as “popularizing” RLHF. In that paragraph, we also cited Chambers and Echenique (2016), Bennet et al. (2007), Knox and Stone (2008), and Wirth et al. (2017). We previously wrote things to lean heavily on the Wirth et al. (2017) survey of preference-based RL methods. But we agree with doing more to describe pre-2017 work. **We have now added [Knox and Stone (2010)](https://www.cs.utexas.edu/~ai-lab/pubs/AAMAS10-knox.pdf), [Knox and Stone (2012)](https://www.cs.utexas.edu/users/ai-lab/pubs/AAMAS12-knox.pdf), [Akrour et al. (2011)](https://link.springer.com/chapter/10.1007/978-3-642-23780-5_11), [Akrour et al. (2012)](https://link.springer.com/chapter/10.1007/978-3-642-33486-3_8), [Furncranz et al. (2012)](https://link.springer.com/article/10.1007/s10994-012-5313-8), and [Griffith et al. (2013)](https://proceedings.neurips.cc/paper_files/paper/2013/hash/e034fb6b66aacc1d48f445ddfb08da98-Abstract.html) to the third introduction paragraph. In the discussion section, we removed one mention of Christiano et al. (2017) and added references to [Lewis (1990)](https://www.tandfonline.com/doi/abs/10.1080/07370024.1990.9667152), [Ng et al. (1999)](https://people.eecs.berkeley.edu/~pabbeel/cs287-fa09/readings/NgHaradaRussell-shaping-ICML1999.pdf), and [Sutton and Barto, 1999](https://www.cell.com/trends/cognitive-sciences/fulltext/S1364-6613(99)01331-5) as examples of how the components of RLHF have roots in research from before 2000.** We are open to adding others as well.
> - Concerning related literature, the paper discussed multiple works on human in the loop-RL in Section 4.2.2 (e.g. Macglashan et al., 2017); IRL in Section 4.2.1 (e.g. Ng et al. 2000); offline RL in Section 4.1.2 (e.g. Levine et al., 2020); and uncertainty estimates in Section 4.2.2 (e.g. Gleave and Irving, 2022). Are there others you would recommend adding?
> 3. Re: Tractable/fundamental distinction – Thank you for the advice on the tractable/fundamental distinction. Our motivation behind it is to distinguish between problems that can be addressed with improvements to RLHF from problems that need to be addressed with non-RLHF approaches. We think that this is a useful point to make because RLHF is not equipped to handle certain problems. **We updated the paragraph in section 3 where we introduce this distinction to explain why we do this instead of simply what we do.** As you mentioned, we provide explanations of each classification as tractable/fundamental in the Appendix, but we are open to reconsidering any specific ones. We are also open to using different words other than “tractable” and “fundamental.” Would you recommend any other vocabulary?
> 4. **We agree that some solutions like IRL are best viewed as old techniques to supplement RLHF instead of new ones to replace it. We updated our discussion in section 4 to explain this.** Regarding IRL specifically, we think it is an open question how useful IRL might be with LLMs. Given the demonstrated efficacy of supervised finetuning the policy on good demonstrations (e.g. instruction finetuning), it seems plausible that reward models could efficiently learn from demonstrations as well. We also added a reference to some recent work from [Li et al. (2023)](https://arxiv.org/abs//2310.11589) which relates to this. Please let us know if there are other things that you would recommend to do about this. Thanks!

---

> > ### Author Response · Authors · 2023-10-27
> > **Response to XNWD + updates part II**
> >
> > Replies to minor items in order
> >
> > 1. Re: data collection leading to harmful biases – We equate distribution shift to a form of bias (e.g. overrepresenting some datapoints and underrepresenting others). But we see the problem in how “biases” can mean multiple things. We rewrote the paragraph to fix this by explaining the link between a restrictive training distribution and poorly-representative outcomes.
> > 2. Thanks! We agree with calling it a cost/quality/quantity tradeoff. We also agree that it is not clear the extent to which additional finetuning data can help. We added both of these points.
> > 3. We use “reward model” to refer to the system that is trained to administer reward instead of an abstract model of the process by which human feedback is used to train the model. We updated the first paragraph of section 3.2 to refer to the reward model as something that is “trained” to clarify this. We agree that the limitations of feedback types are at the intersection of the feedback process and reward model. However, after discussing this, we lean toward not moving it to section 3.2 because addressing the issue involves changing the feedback process. This can happen before the reward model architecture is selected and initialized.
> > 4. We are unsure about this point. Could you elaborate? Consider an example with two annotators with some preferences that fundamentally conflict. The feedback process could represent both of them by simply gathering data from them separately. But a single reward model cannot incentivize behavior from a policy that will satisfy both of them at the same time. This is our reasoning for putting this in section 3.2.
> > 5. We agree that misgeneralization is not specific to reward modeling. We also discuss policy misgeneralization in 3.3.2 and make the observation in Section 6 that many of the practical problems with RLHF are old ones that are common to many ML applications. To be more thorough, we opted to include these problems even thought they are indeed old.
> > 6. See above. We agree about offline RL having well-known challenges, but our goal is thoroughness.
> > 7. We lengthened the discussion of power seeking to explain it more thoroughly. Power-seeking policies are generally a concern with agents who have more open-ended option spaces than an LM (e.g. an agentic system seeking more power or compute). However, we provide one type of specification gaming which is an example of power-seeking from LLMs: “For example, a question-answering LLM trained with RLHF has the incentive to learn to influence human interlocutors in order to avoid conversations about challenging topics.” We are open to additional comments on this.
> > 8. Regarding, "The pretrained model introduces biases into policy optimization". This is not referring to the reward model. Here, we are referring to the pretrained policy network. We rewrote this as "The pretrained policy introduces biases into policy optimization."
> > 9. We agree that RL agents could be trained with distribution-matching objectives. This seems similar to the offline-RL work that we cite in the “Aligning LLMs through supervised learning,” paragraph in section 4.2. We added additional mention of this in section 3.3.3 with a pointer to this paragraph.
> > 10. One of the possible solutions to this issue seems to be synchronously training the policy and reward model (e.g. as done in Christiano et al, (2017)) but this is notably not done for SOTA LLMs. This is the main reason why we refer to addressing this as an open problem.
> >
> > Thanks!

---

> > > ### Comment · Reviewer_XNWD · 2023-11-06
> > >
> > > Just to bring a closure to the reviewing process, here are my comments on the major items' responses:
> > > 1- Thank you.
> > > 2- Thank you for RLHF literature. About Offline RL literature, I saw that some papers (mostly surveys) were cited far into the document. I think it should be more central and in more depth. Same for Human-in-the-loop, I felt it was a bit superficial. Maybe, it's because I feel that human-in-the-loop is a wider field than RLHF
> > > 3- Tractable/fundamental distinction – it still feels very artificial to me. I don't want to run into specifics but why is RLHF + anything else not RLHF anymore? How is it different from a tractable solution of a problem, since it will always also involve RLHF + something else. So, again, my opinion is that it brings more confusion than clarity and I would still personally recommend doing without them.
> > > 4- Thank you.

---

> > > > ### Author Response · Authors · 2023-11-06
> > > > **Trying to balance two goals**
> > > >
> > > > Thanks for the reply.
> > > >
> > > > Re: 3 -- We ultimately agree that there are edge cases between our definitions of tractable and fundamental (we acknowledge this for some problems in appendix B). We have found it challenging to work with this distinction because of this.
> > > >
> > > > Some problems just need to be worked on more. For example, "Selecting representative humans and getting them to provide quality feedback is difficult." Other problems are due to more basic limitations of human nature, reward models, or RL. For example, "A single reward function cannot represent a diverse society of humans." We think that there is a distinction between these kinds of challenges that is important. For the former case, we want this paper to help point the research community toward places where progress can clearly be made. For the latter, we want to make a different point about problems that RLHF will always face.
> > > >
> > > > So we agree that it is challenging to try neatly dividing challenges into tractable/fundamental categories. But we also hope to point out how some challenges do meaningfully differ in how addressable they are with improved techniques. Do you think there could be a way to avoid confusion but still make this point?
> > > >
> > > > Minor, re: "why is RLHF + anything else not RLHF anymore?" Consider an example in which someone uses RLHF plus supervised finetuning to train a model. We think it would be misleading to simply say that the method used to train the model was "reinforcement learning from human feedback".
> > > >
> > > > Overall, thanks for the dialogue and thorough review!

---

### Review · Reviewer_PWoB · 2023-10-22

**Summary Of Contributions:**

The paper proposes a survey of challenges in the Reinforcement Learning with from Human Feedback (RLHF) literature, which is currently the state of the art framework for finetuning language models using fairly straightforward human feedback.
The manuscript summarizes the existing literature into a novel taxonomy of problems, clustering issues on whether they are  "tractable" problems within the RLHF framework, or fundamental limitations of the method.

The authors also review a broader set of alignment and AI safety research which discuss and/or is related to RLHF. Finally, the manuscript concludes with a discussion on governance and accountability challenges in the community developing LLMs with LRH, with highlighted pathways and suggested follow-up work on the topic.

**Audience:**

Yes

**Claims And Evidence:**

Yes

**Requested Changes:**

Not much, really. See perhaps point 3 in the weaknesses section above. Frankly, this paper well passes the bar for publication at TMLR.

**Strengths And Weaknesses:**

## Strengths

This is simply an excellent review. By reading the manuscript, the average TMLR community member will be able to go from 0 to having a great view and overall picture of the RLHF literature. I applaud the authors for getting the right balance between coverage and readability. The paper is also optimized for understanding, with many key points being summarized and reiterated consistently.

Really, great job. One of the best surveys in the ML literature I have ever had the pleasure to read and review.

## Weaknesses

1. Focus on alignment is perhaps somewhat controversial. RLHF is fundamentally a general fine tuning technique that uses human-based sparse feedback and an RL loss, and while alignment is one of its main uses, it's not the only one. I wonder if it had been better to take a broader POV on the matter.
2. It feels this review could have been made tighter by splitting into 3 manuscript: one about the algorithmic limitations and challenges of RLHF, another about the broader issues arising from collecting data from humans, and a final one about auditing and accountability. I'm not sure whether putting all 3 together fully serves the reader and the authors to advance all these narratives, but I do appreciate a good position paper, so I'm not sure myself either way.
3. (nit) The data quality section (3.1.3) includes a fundamental limitation that seems to be more related to section 3.1.4 -- I wonder if that was an intentional separation, or perhaps an editing mistake? Signal / cost / efficiency seem to be sufficently linked that it might be worth rethinking the current categorization.

---

> ### Author Response · Authors · 2023-10-27
> **Response to PWoB**
>
> Thank you! We appreciate the feedback, and we are glad that you enjoyed reading the paper and found it to be one of the best surveys that you have seen in the literature! We are happy that you think it well-passes the bar for publication.
>
> We have posted an update to the paper and describe new changes to it in our reply to you and the other two reviewers.
>
> Replies to weaknesses in order:
>
> 1. It has been a bit of a challenge to balance a focus on RLHF in general with an emphasis on its most prominent application: aligning state-of-the-art LLMs. One thing we think can help to at least avoid confusion is to add to section 2 a brief discussion of non-LLM uses of RLHF.
> 2. We are glad if you think that this paper could have been multiple, and we agree that this survey has some qualities of a position paper. Currently, our plan will be to continue with the TMLR process, but we will discuss reorganizing things if it is not accepted. Meanwhile, do you think there is anything we can change in the abstract to help lay out the content better for the reader?
> 3. Thank you for this point! We see the connection here. There are multiple types of tradeoffs. Some involve the feedback type (e.g. preference feedback is easy but gives limited info; language feedback is rich but high effort) and some involve the overall process (sourcing data from product users versus trained experts). We added to Section 3.1.4 to explain this contrast.

---

### Decision · Action_Editor_92Zz · 2023-12-01

**Recommendation:** Accept as is

**Comment:**

The paper contributes a comprehensive survey and analysis of challenges in Reinforcement Learning from Human Feedback (RLHF) for fine-tuning large language models (LLMs), proposing solutions and research directions.
All the reviewers agree that this paper provides a thorough literature review on RLHF and deserves publication.
On the other hand, there were concerns about the lack of a distinct technical contribution and the need for improvements in the background section. The authors have addressed these concerns in their rebuttals and have incorporated the suggested changes.
The paper can be accepted for publication. Congratulations!

**Audience:**

Yes

**Claims And Evidence:**

Yes